# Two-photon calcium imaging of the medial prefrontal cortex and hippocampus without cortical invasion

**Masashi Kondo[1,2], Kenta Kobayashi[3], Masamichi Ohkura[4], Junichi Nakai[4], Masanori Matsuzaki[1,2]\***

[1]Department of Physiology, Graduate School of Medicine, The University of Tokyo, Tokyo, Japan; [2]Division of Brain Circuits, National Institute for Basic Biology, Okazaki, Japan; [3]Section of Viral Vector Development, National Institute for Physiological Sciences, Okazaki, Japan; [4]Brain and Body System Science Institute, Graduate School of Science and Engineering, Saitama University, Saitama, Japan

**Abstract** In vivo two-photon calcium imaging currently allows us to observe the activity of multiple neurons up to ~900 μm below the cortical surface without cortical invasion. However, many important brain areas are located deeper than this. Here, we used an 1100 nm laser that underfilled the back aperture of the objective together with red genetically encoded calcium indicators to establish two-photon calcium imaging of the intact mouse brain and detect neural activity up to 1200 μm from the cortical surface. This imaging was obtained from the medial prefrontal cortex (the prelimbic area) and the hippocampal CA1 region. We found that neural activity before water delivery repeated at a constant interval was higher in the prelimbic area than in layer 2/3 of the secondary motor area. Reducing the invasiveness of imaging is an important strategy to reveal the intact brain processes active in cognition and memory.

DOI: https://doi.org/10.7554/eLife.26839.001

**\*For correspondence:**
mzakim@m.u-tokyo.ac.jp

**Competing interests:** The authors declare that no competing interests exist.

## Introduction

Two-photon calcium imaging reveals the in vivo activity of multiple neurons at cellular and subcellular resolution (*Jia et al., 2010*; *Ohki et al., 2005*). Recent work demonstrates that by exciting red-fluorescent calcium indicators with a laser at wavelengths of 1000–1100 nm through a cranial window, it is possible to image neural activity in the mouse sensory cortex at depths of 800–900 μm from the cortical surface (corresponding to layers 5 and 6) (*Dana et al., 2016*; *Tischbirek et al., 2015*). However, for functional imaging of deeper regions such as the medial prefrontal cortex, hippocampus, and basal ganglia, invasive penetration is unavoidable; it is necessary to insert a microlens or a microprism into the cortical tissue, or to remove the cortical tissue above the target region (*Attardo et al., 2015*; *Dombeck et al., 2010*; *Low et al., 2014*; *Pilz et al., 2016*). The difficulty of deep imaging is mainly caused by refractive index mismatch and light scattering within the tissue (*Helmchen and Denk, 2005*; *Tung et al., 2004*). These can be weakened when objectives with low numerical aperture (NA) are used because the angle of light emitted from the objective is smaller and the light-path length within the tissue is shorter than when high NA objectives are used. However, when low NA objectives are used, the spatial resolution and collection efficiency of emitted fluorescent signals are worse than when high NA objectives are used. If a high NA objective is used in conjunction with underfilling of the back aperture by the excitation laser, the collection efficiency of the fluorescent signal remains high. If the effective NA for the excitation light is small but sufficient to resolve single neurons (10–15 μm along the Z axis; *Lecoq et al., 2014*), this technique may increase the maximal depth for two-photon calcium imaging of neuronal somata. Although this

**eLife digest** Microscopes can now reveal what individual cells are doing inside a living brain. In a technique called two-photon microscopy, light-sensitive proteins are introduced into the brain cells. A laser then shines light of a specific wavelength into the brain. Whenever one of the proteins in an active brain cell absorbs some light from the laser, it gives off light that a sensor can detect.

Yet, a two-photon microscope could only "see" up to about 900 micrometers from the brain's surface. This is because light scatters as it travels through brain tissue. Shorter wavelengths scatter the most; so two-photon microscopes use infrared lasers, which have a longer wavelength than visible light. Even so, structures deeper within the brain like the hippocampus and medial prefrontal cortex remained out of range. The only way to see these structures – which are involved in memory and planning – was to damage the brain by inserting a lens or by removing the overlying tissue. But such damage may also change brain activity.

Kondo et al. have now found a way to image brain cells up to 1,200 micrometers below the surface of an intact mouse brain. The new approach uses an optimized microscope and a laser that generates even longer wavelength light. It also makes use of proteins that give off red light, rather than yellow or green. These changes made it possible to view activity in the medial prefrontal cortex and hippocampus. The brain cells showed no signs of damage after about 30 minutes of viewing. This suggests that the approach does not cause overheating or kill cells.

Many questions remain about what happens deep within an active brain. By allowing neuroscientists to follow the activity of brain cells over months, for example as an animal learns a task, these improvements to two-photon microscopy could lead to new insights into the processes of learning and decision-making. Kondo et al. hope that other researchers will find more ways to use the refined technique in their own experiments.

DOI: https://doi.org/10.7554/eLife.26839.002

technique has been theoretically predicted and partially demonstrated in the skin (*Helmchen and Denk, 2005*; *Tung et al., 2004*), it has not been applied to deep imaging of neural activity in behaving animals. In addition, long-wavelength excitation light and red-fluorescent genetically encoded calcium indicators (red GECIs; *Dana et al., 2016*; *Inoue et al., 2015*; *Ohkura et al., 2012*) are suitable for deep imaging because light scattering is weaker at longer wavelengths. Here, we demonstrate that by underfilling high NA objectives to reduce the effective NA for excitation to approximately 0.5 and exciting red GECIs with an 1100-nm laser, we could detect the activity of multiple neurons in the medial prefrontal cortex (the prelimbic [PL] area) and the hippocampal CA1 region at depths of 1.0–1.2 mm in behaving mice without the need for invasive penetration or removal of cortical tissue.

## Results

To reduce the effective NA, the back aperture of a high NA (1.00) objective was underfilled with a diameter-narrowed 1100 nm laser beam ($1/e^2$-width was 7.2 mm, compared with the back aperture of 14.4 mm) (*Figure 1A*; *Matsuzaki et al., 2008*). The effective NA was calculated to be roughly 0.5 (see Materials and methods). To examine the effect of underfilling the objective on spatial resolution, 2-µm-diameter fluorescent beads embedded in 2% agarose were imaged through the same glass window used for in vivo GECI imaging (approximately 0.77 mm total thickness). When the objective was underfilled, the full-widths at half-maximum (FWHMs) were $2.28 \pm 0.05$ µm (mean ± s.d., $n = 5$ beads) laterally and $6.95 \pm 0.13$ µm ($n = 5$ beads) axially (*Figure 1Bi*). These values were greater than the FWHMs of $2.15 \pm 0.05$ µm ($n = 4$ beads) laterally and $4.37 \pm 0.06$ µm ($n = 4$ beads) axially when the objective was overfilled (*Figure 1Bii*), but comparable with those used for two-photon calcium imaging of multiple neurons with cellular resolution (*Lecoq et al., 2014*; *Sadakane et al., 2015*; *Stirman et al., 2016*).

Next, we examined whether underfilling the objective was effective for deep imaging in the mouse cerebral cortex. Adeno-associated virus (AAV) carrying the tdTomato gene was injected into the intact medial frontal cortex (mFrC) of 2- to 3-month-old mice. Two to three weeks post-injection, we imaged tdTomato-expressing neurons in the mFrC at depths of 100–1200 µm from the cortical

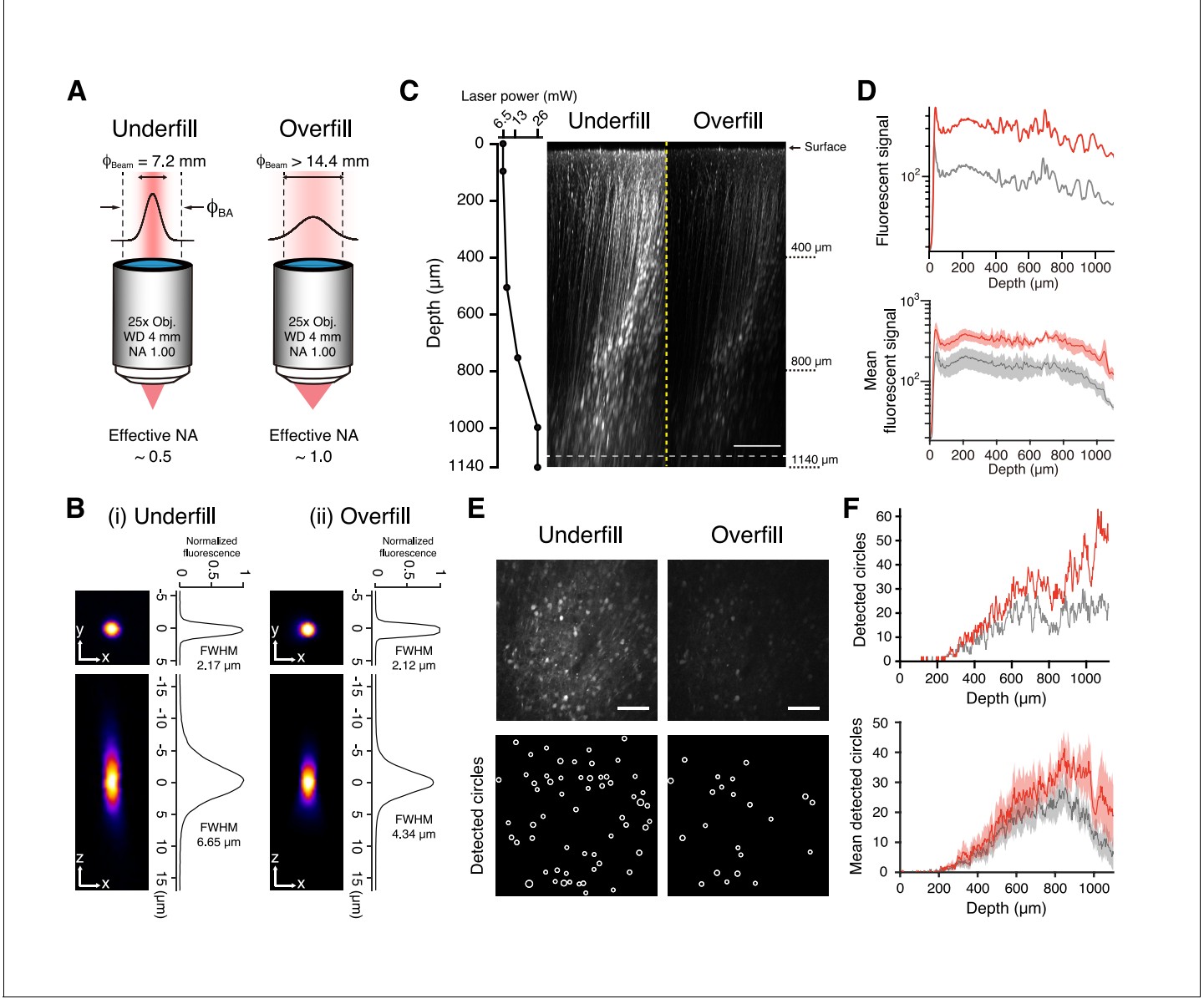

**Figure 1.** Two-photon imaging with underfilled and overfilled objectives. (**A**) Schematic illustration of an objective with NA of 1.0, magnification of 25, and working distance of 4 mm, which was underfilled (left) and overfilled (right) with an excitation laser. 'Underfill' and 'Overfill' denote that $1/e^2$-width of the excitation laser beam, $\phi_{Beam}$, is narrower and wider than the back aperture of the objective, $\phi_{BA}$, respectively. The calculated effective NAs are indicated below. (**B i, ii**) Representative XY (left top) and XZ (left bottom) images of a 2 µm fluorescence bead and their profile (right) when the objective was underfilled (i) and overfilled (ii). (**C**) Representative XZ images of mFrC expressing tdTomato in the underfilled and overfilled configurations (right images). The laser power was determined at six depth points (dots) and gradually increased between dots as the depth increased (left graph). The power at each cortical depth was equal in both underfilled and overfilled configurations. Contrast was not enhanced in either image. The images are the maximum intensity projection of XYZ images along the Y axis. Scale bar, 200 µm. (**D**) Top, Z profiles of the mean fluorescence signals from the brightest pixels from the same volume as in (**C**) in the underfilled (red) and overfilled (gray) configurations. Bottom, Z profiles of fluorescence signals averaged over four volumes from two mice. Light shading indicates the s.e.m. (**E**) XY images in the underfilled (left) and overfilled (right) configurations at the depth indicated by the dotted line in (**C**), 1060 µm from the brain surface. Contrast enhancement was not performed in these images. Bottom, spatial distributions of circles detected by the Hough transform method. Scale bar, 100 µm. (**F**) Top, Z profiles of the number of detected circles from the same volume as in (**C**) in the underfilled (red) and overfilled (gray) configurations. Bottom, Z profiles of the number of detected circles averaged over four volumes from two mice. Light shading indicates the s.e.m.

DOI: https://doi.org/10.7554/eLife.26839.003

The following source data is available for figure 1:

**Source data 1.** Data of bright fluorescent signals and the number of detected circles at all depths in four fields in underfilled and overfilled configurations.
DOI: https://doi.org/10.7554/eLife.26839.004

surface in anesthetized and head-restrained mice through a cranial window in underfilled and over-filled configurations (*Figure 1C* and *Video 1*). We adjusted the laser power at the front aperture of the objective such that it was equal in both configurations. For all depths, the mean bright fluorescent signal (*Kobat et al., 2009*), which was assumed to reflect the fluorescence from tdTomato-expressing neurons, was higher in the underfilled than in the overfilled configuration (*Figure 1D,E* and *Figure 1—source data 1*). The number of circles detected by the Hough transform method (see Materials and methods), which is assumed to reflect the number of fluorescent neuronal somata, was similar between the two configurations at depths of 200–600 μm from the cortical surface (*Figure 1F,G* and *Figure 3—source data 1*). However, the number of circles detected at depths > 600 μm was larger in the underfilled than in the overfilled configuration (*Figure 1E*). Thus, underfilling the objective was more effective at detecting neuronal morphology at depths > 600 μm.

Next, we determined whether underfilling the objective allowed us to detect neural activity at depths > 900 μm from the cortical surface. Three to four weeks after an injection of AAV carrying the R-CaMP1.07 gene (*Ohkura et al., 2012*) into the intact mFrC of 2- to 3-month-old mice, we observed R-CaMP1.07-expressing neurons in the mFrC at depths of 100–1200 μm in awake and head-restrained mice (*Figure 2A–C* and *Video 2*). Using laser power of 170–180 mW at the front aperture of the objective, we detected calcium transients at depths of 1.0–1.2 mm (*Figure 2D,E*).

This laser power was higher than that used for two-photon imaging of cortical layers 5 and/or 6 (the maximum power used was: 150 mW in *Dana et al., 2016*; 114 mW in *Masamizu et al., 2014*; 170 mW in *Tischbirek et al., 2015*). Therefore, we examined whether imaging deep mFrC with a 180 mW, 1100 nm laser caused inflammation and heating-induced responses. We used anti-Glial Fibrillary Acidic Protein (GFAP) as a marker for activated astrocytes, anti-Ionized calcium binding adapter molecule 1 (Iba1) as a marker for activated microglia, and anti-Heat Shock Protein 70/72 (HSP70/72) as a marker for heat-induced responses in glial cells and neurons (*Figure 3*; *Podgorski and Ranganathan, 2016*). Immunostaining intensity was quantified under five conditions: without imaging (as a negative control), after 15-min imaging at 800–900 μm depths, after 30-min imaging at 900–1100 μm depths, after 30-min imaging at 300–400 μm depths (as a positive control), and after 30-min imaging at 200–300 μm depths, using an objective overfilled with a 200 mW, 920 nm laser and a slow scanning mode (as a strong positive control). The immunoreactivity after 15- or 30-min imaging at 800–1100 μm depths was not different from that in negative control mice (*Figure 3*, *Figure 3—figure supplement 1*, and *Figure 3—source data 1*). By contrast, imaging at 300–400 μm depths with an 1100 nm laser caused significantly higher anti-GFAP and anti-Iba1 immunoreactivity than deep imaging (*Figure 3D–F* and *Figure 3—source data 1*). Imaging at 200–300 μm depths with a 920 nm laser in the slow scanning mode caused significantly higher immunoreactivity of all antibodies

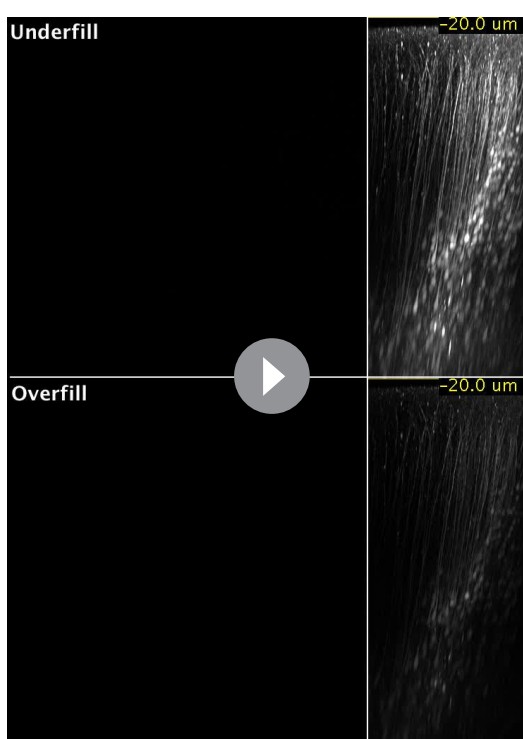

**Video 1.** Representative two-photon XYZ images of the mFrC in a tdTomato-expressing animal with underfilled and overfilled objectives. The depth increment in the image stack was 2.5 μm, and the lowest imaging depth was 1120 μm from the cortical surface. The field of view is 509.12 μm × 509.12 μm (512 × 512 pixels). Each image is the average of 16 frames acquired by resonant scanning at 30 Hz. The mouse was anesthetized. The movie was denoised with a spatial Gaussian filter (σ = 0.5). The upper and lower half images correspond to the underfilled and overfilled configurations, respectively. The right image corresponds to the XZ plane of the XYZ images (maximum intensity projection toward the Y dimension) and the horizontal yellow line indicates the depth of each left XY image in each configuration.

DOI: https://doi.org/10.7554/eLife.26839.005

than deep imaging (*Figure 3D–F*, *Figure 3—figure supplement 2*, and *Figure 3—source data 1*). These results indicate that our immunostaining assays are sensitive enough to detect laser-induced tissue damage and are consistent with a study reporting that heat-induced cell responses occur when two-photon imaging is performed at a 250 μm depth with $\geq\sim300$ mW laser power at 920 nm (*Podgorski and Ranganathan, 2016*). However, 15–30 min two-photon imaging of the deep area with an 1100 nm laser at 180 mW power did not cause any apparent histological injury.

We also examined whether neural activity was affected by deep imaging. In a 25-min continuous imaging session at 900–1100 μm depths in awake mice, we compared the activity of imaged neurons in the first 5-min period with that in the last 5-min period. There was no difference in the mean inferred activities (by a constrained non-negative matrix factorization algorithm; *Pnevmatikakis et al., 2016*) between the two periods (*Figure 3—figure supplement 3*). This indicates that two-photon imaging of the deep area with ~180 mW laser power did not appear to alter neural excitability.

To demonstrate the utility of this method for identifying neural functions in deep areas in the intact brain, we examined neural activity in the mFrC over ~1 mm depth during simple conditioning. Head-restrained mice were conditioned to the delivery of a drop of water with an inter-delivery interval of 20 s (*Figure 4A*). As each session progressed (one session per day), the licking response rate to water delivery increased and licking became faster (*Figure 4A–D*). From the fourth–fifth sessions onwards, we performed two-photon calcium imaging of the mFrC at cortical depths of 100–1200 μm (*Video 3*). The imaging fields were classified into three areas according to depth (*Paxinos and Franklin, 2007*): the superficial area (100–300 μm, corresponding to layer 2/3 in the secondary motor area, M2), the middle area (300–800 μm, corresponding to layer 5 in M2), and the deep area (800–1200 μm, roughly corresponding to layer 6 in M2 and the PL area). In all three areas, approximately 50% of neurons showed a peak in the mean (trial-averaged) activity during the 5 s after water delivery (*Figure 4E,F* and *Figure 4—figure supplement 1*), which was presumably related to licking and water acquisition (*Figure 4B*). Additionally, approximately 30% of neurons in all three areas showed a peak in the mean activity during the 10 s before water delivery (pre-reward period; *Figure 4E,F* and *Figure 4—figure supplement 1*). The sequential distribution of the times of peak activity was not an artifact of ordering the neurons according to the time of peak activity, as the ratio of the mean activity around the peak activity to the baseline activity (ridge-to-background ratio; *Harvey et al., 2012*) was significantly higher than that of shuffled data (*Figure 4G,H*). Additionally, the sequential distribution of neurons with peak activity during the pre-reward period was not an artifact (*Figure 4—figure supplement 2*). As the mFrC demonstrates strong activity before movement starts (*Friedman et al., 2015*; *Kim et al., 2016a*; *Pinto and Dan, 2015*; *Sul et al., 2011*), we focused on the activity during the pre-reward period. When 5 s windows were chosen from the pre-reward period, the ridge-to-background ratios of deep area neurons with peak activity during each 5 s window were frequently higher than those in the shuffled data (*Figure 4—figure supplement 3*). To determine whether the activity pattern across trials was stable for individual neurons with peak activity during the pre-reward period, we calculated the correlation coefficient between the times of peak activity of two randomly separated groups of trials (*Figure 4—figure supplement 4A,B*; see details in Materials and methods section) and found that it was higher in the deep area than in the superficial area (*Figure 4—figure supplement 4C*). This indicates that the PL neurons reliably code the neural activity during the pre-reward period.

In addition to the mPrC, we examined whether neural activity in the hippocampus can be imaged without removal of the neocortical tissue lying above it (*Figure 5A*). CA1 GFP-expressing neurons can be detected by two-photon microscopy in 4-week-old mice, but not in 6- to 9-week-old mice (*Kawakami et al., 2013*). Therefore, we injected AAV-jRGECO1a (*Dana et al., 2016*) into the hippocampus of mice aged between 12 and 14 days, and then performed imaging after another 2 weeks. For imaging, the 15.1 mm back aperture of the objective (NA 1.05) was underfilled with a 7.2 mm laser beam (*Figure 5A*). When we deepened the focal plane below the white matter to depths of 900–1000 μm, we observed densely distributed fluorescent neurons typically located in the CA1 pyramidal layer (*Figure 5B,C* and *Video 4*), as described previously (*Dombeck et al., 2010*), and clearly detected spontaneous calcium transients from these neurons (*Figure 5C*). No cell death or strong damage was apparent after 15 min of imaging (*Figure 5D,E*). By contrast, we could not detect any neural morphology or activity in the CA1 region of the mice when they were 3 months old. In the present surgery schedule, the period used for reward delivery conditioning and imaging

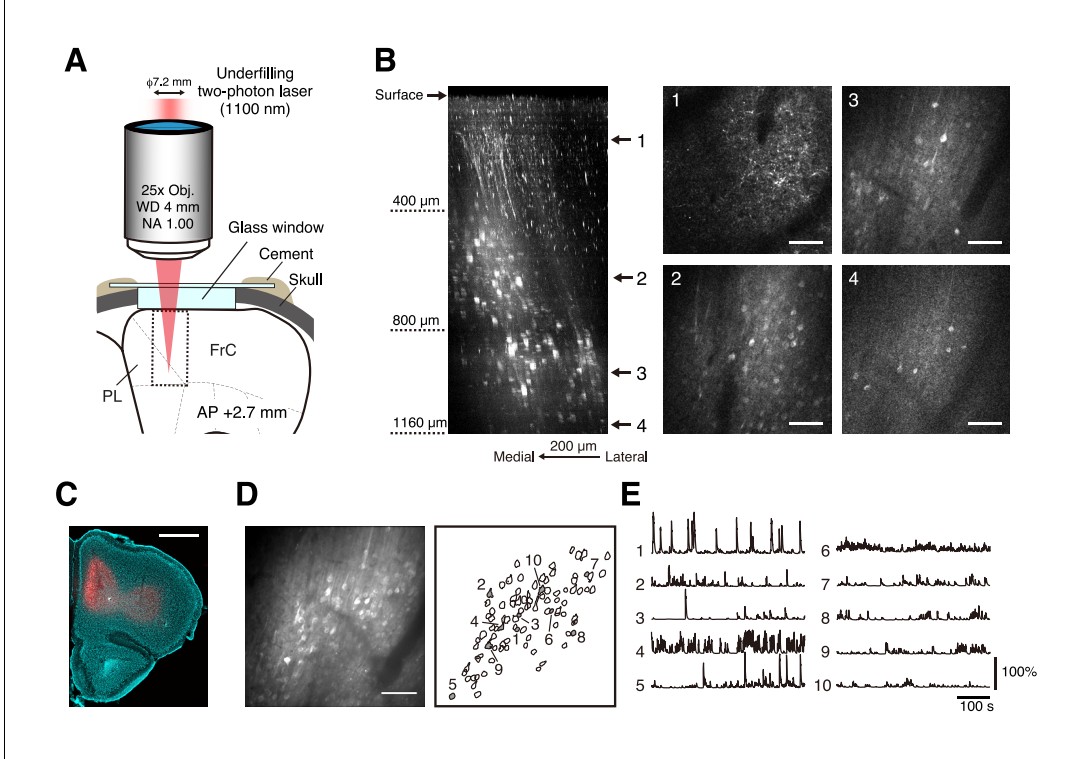

**Figure 2.** Two-photon calcium imaging of intact medial frontal cortex. (**A**) Schematic illustration of in vivo two-photon imaging of the mFrC. The dotted square indicates the location at which the Z-stack images in (**B**) were acquired. Imaging stage was tilted so that the glass window was perpendicular to the optical axis (see Materials and methods). FrC, frontal cortex; PL, prelimbic area. Immersion water was placed between the objective and the cranial window, but is not illustrated for visual clarity. (**B**) Left, representative XZ image of mFrC expressing R-CaMP1.07. The images are maximum intensity projections of XYZ images along the Y axis. These were acquired in awake mice, and the multiple horizontal dark lines are motion artifacts. Scale bar, 200 μm. Middle and right, four XY images at the depths indicated by the number at the left. Scale bar, 100 μm. (**C**) Expression of R-CaMP1.07 (red) and cell nuclei (NeuroTrace 435, cyan) after imaging 11 fields at depths of 340–1140 μm for 5 days. The total duration of imaging was 200 min. Scale bar, 1 mm. (**D**) Left, representative time-averaged XY image of mFrC expressing R-CaMP1.07 at a depth of 1030 μm. Scale bar, 100 μm. Right, spatial distribution of neurons identified by the constrained non-negative matrix factorization (cNMF) algorithm. (**E**) Calcium transient traces for the numbered filled contours are shown in (**D**). These measurements were taken while the animal was quiet, more than 15 min after the conditioning experiment.

DOI: https://doi.org/10.7554/eLife.26839.006

was shorter, and the young mice were conditioned more slowly than the adult mice used for imaging of the mFrC (compare the middle and right panels in *Figure 6A* and *Figure 4B,C*). In the fourth conditioning session, we conducted two-photon calcium imaging of hippocampal CA1 neurons (*Video 5*). The majority of active neurons showed peak activity 0–5 s after water delivery (*Figure 6B*). The time of peak activity of neurons showing peak activity during the pre-reward period was not stable across trials (*Figure 6C,D*).

## Discussion

Here, we demonstrated that underfilling the objective was effective for deep (600–1200 μm from the cortical surface) imaging of neural morphology and activity in the mouse brain over the course of several days. This may be due not only to reduced light scattering but also to the high fluorescent signal in the underfilled configuration; fluorescence is integrated over a larger focal volume in the underfilled configuration than in the overfilled configuration. Deep imaging of neural activity required a relatively high-power laser. However, we confirmed that 15- and 30-min imaging did not produce any apparent morphological or functional damage to the brain tissue. This might be because the excitation (or density of photons) at the focal center is lower in the underfilled configuration than in the overfilled configuration (*Helmchen and Denk, 2005*) and the heat derived from the 1100 nm photon absorbance is relatively low (*Hale and Querry, 1973*).

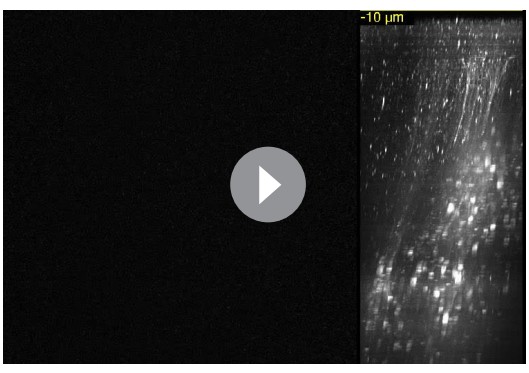

**Video 2.** Representative two-photon XYZ images of the mFrC expressing R-CaMP1.07. The depth increment in the image stack was 2.0 µm, and the lowest imaging depth was 1100 µm. The field of view is 509.12 µm × 509.12 µm (512 × 512 pixels). Each image represents an average of 16 frames acquired by resonant scanning at 30 Hz. The mouse was not anesthetized. Motion correction was not conducted. The movie was denoised with a spatial Gaussian filter (σ = 0.6). The right image corresponds to the XZ plane of the XYZ images (maximum intensity projection toward the Y dimension) and the horizontal yellow line indicates the current depth of the left XY image.
DOI: https://doi.org/10.7554/eLife.26839.007

We could not image the infralimbic area at >1200 µm depths or the hippocampus in adult mice. Adult hippocampus is difficult to image because, as the mouse becomes older, the myelination of the white matter increases (*Bockhorst et al., 2008*). A recent study demonstrated that three-photon calcium imaging with a 1300 nm laser with a high power per pulse (~60 nJ) can access adult hippocampal CA1 neurons (*Ouzounov et al., 2017*). This suggests that light at around 1100 nm may also penetrate the highly myelinated white matter if a laser with higher average power or higher power per pulse (*Kawakami et al., 2015*) is used (the power per pulse in this study was 2.3 nJ). In addition, reduction of the effective NA to ~0.35 (corresponding to an axial resolution of ~10 µm; *Lecoq et al., 2014*; *Stirman et al., 2016*), introduction of adaptive optics to compensate for light scattering (*Ji et al., 2010*), and further improvement of the signal-to-noise ratio of red GECIs (*Dana et al., 2016*; *Inoue et al., 2015*) will certainly be helpful for imaging multicellular activity in the infralimbic area and the hippocampus in adult intact mice. The major disadvantage of reducing the effective NA is a decrease in spatial resolution. In this study, the axial resolution was approximately 7 µm, which is sufficient to resolve single neurons and is unlikely to cause cross-talk between neurons closely located along the Z axis (*Lecoq et al., 2014*). However, if dendritic branches and spines, and axonal branches and boutons, are the imaging target, then the spatial resolution may not be sufficient. When adaptive optics are used, YFP-expressing dendritic spines can be resolved at a depth of 600 µm (*Wang et al., 2015*). Thus, for deep imaging of subcellular activity, a slightly underfilled objective (with an effective NA of ~0.7) combined with adaptive optics may be useful.

We found that neural activity during the pre-reward period was more robust in the deep area (the PL area) than in the superficial area of the mFrC (M2). The PL area is strongly related to the processing of motivation-, attention-, and reward-related information (*Friedman et al., 2015*; *Kim et al., 2016b*; *Otis et al., 2017*; *Pinto and Dan, 2015*), whereas M2 is strongly related to action selection and motor planning (*Li et al., 2015*; *Sul et al., 2011*). Thus, PL activity during the pre-reward period might reflect the expectation of reward delivery, including motivation, reward prediction, or attention to the timing of the water delivery. However, the conditioning in this study did not require any change in mouse brain state before water delivery. Conducting deep imaging during decision-making tasks will help us to understand the hierarchical and/or parallel processing occurring across the PL and M2 areas during decision-making and action.

In the intact brain, it is easy to change the field of view parallel to the cortical surface. An 8-mm-wide glass window can be used for long-term imaging of the whole dorsal neocortex in the mouse (*Kim et al., 2016a*). Objectives with wide fields of view (>3 mm) developed for two-photon imaging (*Sofroniew et al., 2016*; *Stirman et al., 2016*; *Tsai et al., 2015*) can cover the mFrC and the neocortex lying above the hippocampus. The hippocampus connects the mFrC through the thalamus and the entorhinal cortex (*Jin and Maren, 2015*; *Varela et al., 2014*) and is thought to associate spatial, temporal, and reward information, which are required for goal-directed decision-making (*Wikenheiser and Schoenbaum, 2016*). Here, the pre-reward activity in hippocampal CA1 neurons was not stable, likely because the conditioning period in the young mice was not sufficient to form such activity. If the adult hippocampus can be imaged through a wide-field cranial window and objective, the neural activity in both areas could be imaged simultaneously. Deep and wide-field

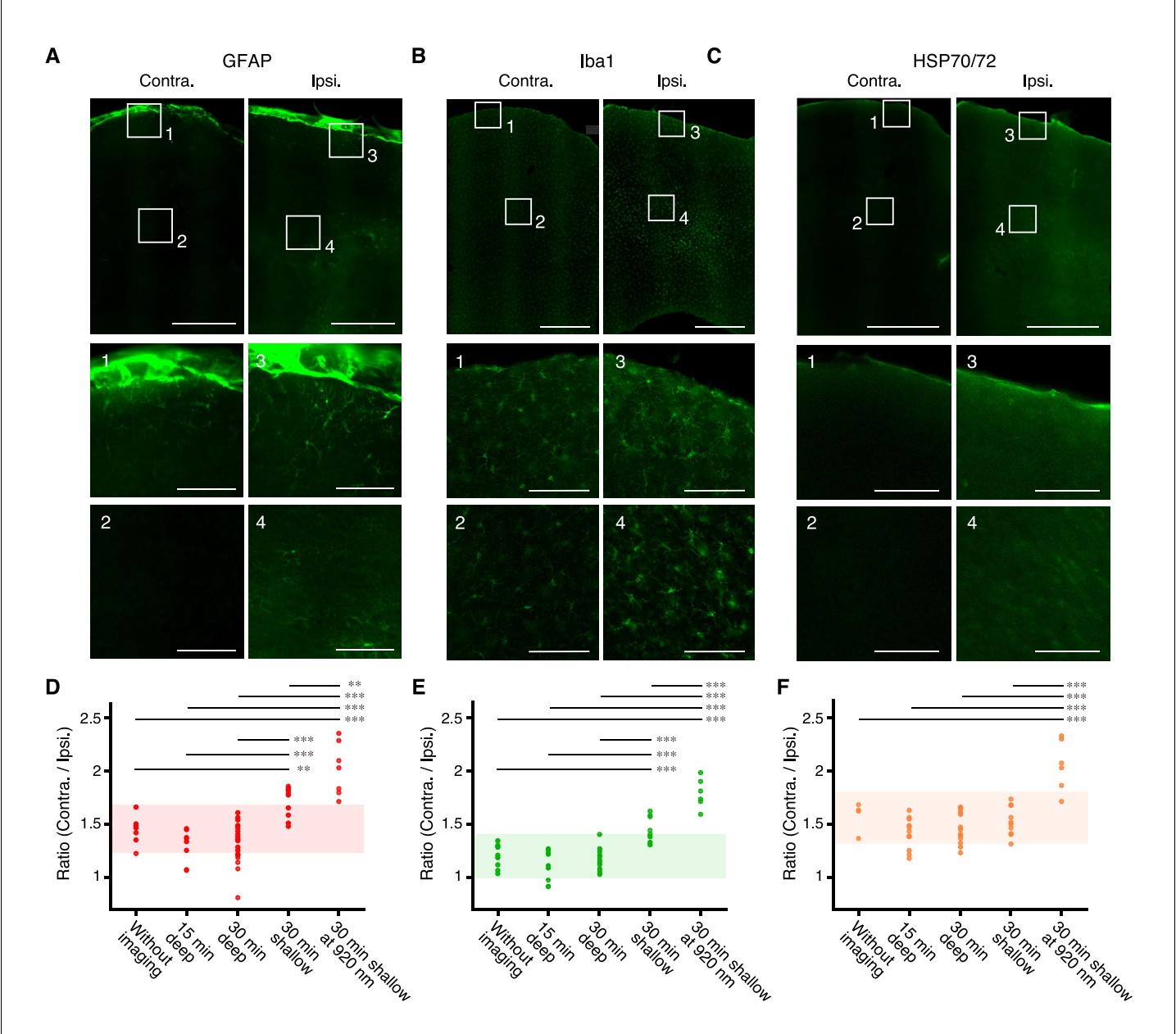

**Figure 3.** Immunoreactivity after deep imaging of the medial frontal cortex. (**A–C**) Top, representative expression of GFAP (**A**), Iba1 (**B**), and HSP70/72 (**C**) in contralateral (left) and imaged (right) hemispheres. Scale bar, 500 µm. Middle and bottom, expanded images of the numbered boxes, including the cortical surface (middle) and imaging sites (bottom). Scale bar, 100 µm. (**D–F**) Ratios of expression intensity for GFAP (**D**), Iba1 (**E**), and HSP70/72 (**F**) in the AAV-injected hemisphere compared with the contralateral hemisphere. Each dot indicates the ratio in each slice that included both hemispheres. Without imaging: mice with only AAV injections and cranial windows (10 slices from two mice as a negative control for GFAP and Iba1; four slices from one mouse as a negative control for HSP70/72). 15 min and 30 min deep: after 15-min imaging at 800–900 µm depths (eight slices from two mice for GFAP; 10 slices from two mice for Iba1; 15 slices from two mice for HSP70/72) and after 30-min imaging at 900–1100 µm depths (21 slices from four mice for GFAP; 21 slices from four mice for Iba1; 12 slices from two mice for HSP70/72), respectively; 30 min shallow: after 30 min imaging at 300–400 µm depths (eight slices from two mice for GFAP; 10 slices from two mice for Iba1; 10 slices from two mice for HSP70/72); 30 min shallow at 920 nm: after 30-min imaging at 200–300 µm depths (overfilled with 200 mW, 920 nm laser in the slow scanning mode) (seven slices from two mice for GFAP; six slices from two mice for Iba1; six slices from two mice for HSP70/72). Shaded areas indicate the mean ±2 s.d. of the immunoreactivity in the negative control experiment. Horizontal bars indicate significant differences (**: $p < 0.01$, ***: $p < 0.001$, one-way ANOVA with Tukey-Kramer method for *post-hoc* multiple comparisons, see *Figure 3—source data 1*). Pairs without bar were not significantly different (see *Figure 3—source data 1*).
DOI: https://doi.org/10.7554/eLife.26839.008

The following source data and figure supplements are available for figure 3:

*Figure 3 continued on next page*

*Figure 3 continued*

**Source data 1.** Data of immunoreactivity across five conditions and their statistics.
DOI: https://doi.org/10.7554/eLife.26839.012
**Figure supplement 1.** Immunoreactivity in the negative control experiment.
DOI: https://doi.org/10.7554/eLife.26839.009
**Figure supplement 2.** Immunoreactivity in the strong positive control experiment.
DOI: https://doi.org/10.7554/eLife.26839.010
**Figure supplement 3.** Calcium transients of neurons in the deep area before and after 15 min imaging.
DOI: https://doi.org/10.7554/eLife.26839.011

two-photon calcium imaging of the intact brain will substantially aid our understanding of the brain circuits that integrate multimodal information in decision-making.

## Materials and methods

### Animals

All animal experiments were approved by the Institutional Animal Care and Use Committee of The University of Tokyo, Japan (Medicine-P16-012). All mice were provided with food and water *ad libitum* and housed in a 12:12 hr light–dark cycle. The mice were not used for other experiments before this study. Male C57BL/6 mice (aged 2–3 months, SLC, Shizuoka, Japan) were utilized for mFrC imaging. Male and female C57BL/6 mice (aged 12–40 days in the young mice, and 2–3 months in the adult mice; Japan SLC, Shizuoka, Japan) were utilized for the imaging experiments in the hippocampus. For experiments using young mice, pups were weaned at P30, and then group-housed until the imaging window was implanted.

### Virus production

In this study, two red-fluorescent genetically encoded calcium indicators, R-CaMP1.07 (*Ohkura et al., 2012*), jRGECO1a (*Dana et al., 2016*) and a calcium-insensitive red-fluorescent protein, tdTomato, were used. For imaging of R-CaMP1.07, the GCaMP3 DNA of pAAV-human synapsin I promoter (hSyn)-GCaMP3-WPRE-hGH polyA (*Masamizu et al., 2014*) was replaced with R-CaMP1.07 DNA from a pN1-R-CaMP1.07 vector construct (*Ohkura et al., 2012*). rAAV2/1-hSyn-R-CaMP1.07 ($1.3 \times 10^{13}$ vector genomes/ml) was produced with pAAV2-1 and purified as described previously (*Kaneda et al., 2011*; *Kobayashi et al., 2016*). tdTomato was expressed via a 1:1 cocktail of viral solutions including rAAV2/1-CaMKII-Cre ($3.16 \times 10^{10}$ vector genomes/ml) and rAAV2/1-CAG-FLEX-tdTomato ($5.1 \times 10^{12}$ vector genomes/ml). rAAV2/1-CaMKII-Cre, rAAV2/1-CAG-FLEX-tdTomato and rAAV2/1-hSyn-NES-jRGECO1a ($2.95 \times 10^{13}$ vector genomes/ml) were obtained from the University of Pennsylvania Gene Therapy Program Vector Core.

### Surgical procedures

#### mFrC

Mice were anesthetized by intramuscular injection of ketamine (74 mg/kg) and xylazine (10 mg/kg) before an incision was made in the skin covering the neocortex. After the mice were anesthetized, atropine (0.5 mg/kg) was injected to reduce bronchial secretion and improve breathing, and an eye ointment (Tarivid; 0.3% w/v ofloxacin, Santen Pharmaceutical, Osaka, Japan) was applied to prevent eye-drying. Body temperature was maintained at 36–37°C with a heating pad. After the exposed skull was cleaned, a head-plate (Tsukasa Giken, Shizuoka, Japan; *Hira et al., 2013*) was attached to the skull using dental cement (Fuji lute BC; GC, Tokyo, Japan, and Bistite II; Tokuyama Dental, Tokyo, Japan). The surface of the intact skull was coated with dental adhesive resin cement (Super bond; Sun Medical, Shiga, Japan) to prevent drying. An isotonic saline solution with 5 w/v% glucose was injected intraperitoneally after the surgery. Mice were allowed to recover for 1 day before virus injection.

Thirty minutes before surgery for virus injection, dexamethasone sodium phosphate (1.32 mg/kg) was administered intraperitoneally to prevent cerebral edema. Mice were anesthetized with isoflurane (3–4% for induction and ~1% during surgery) inhalation and placed on a stereotaxic frame (SR-5M;

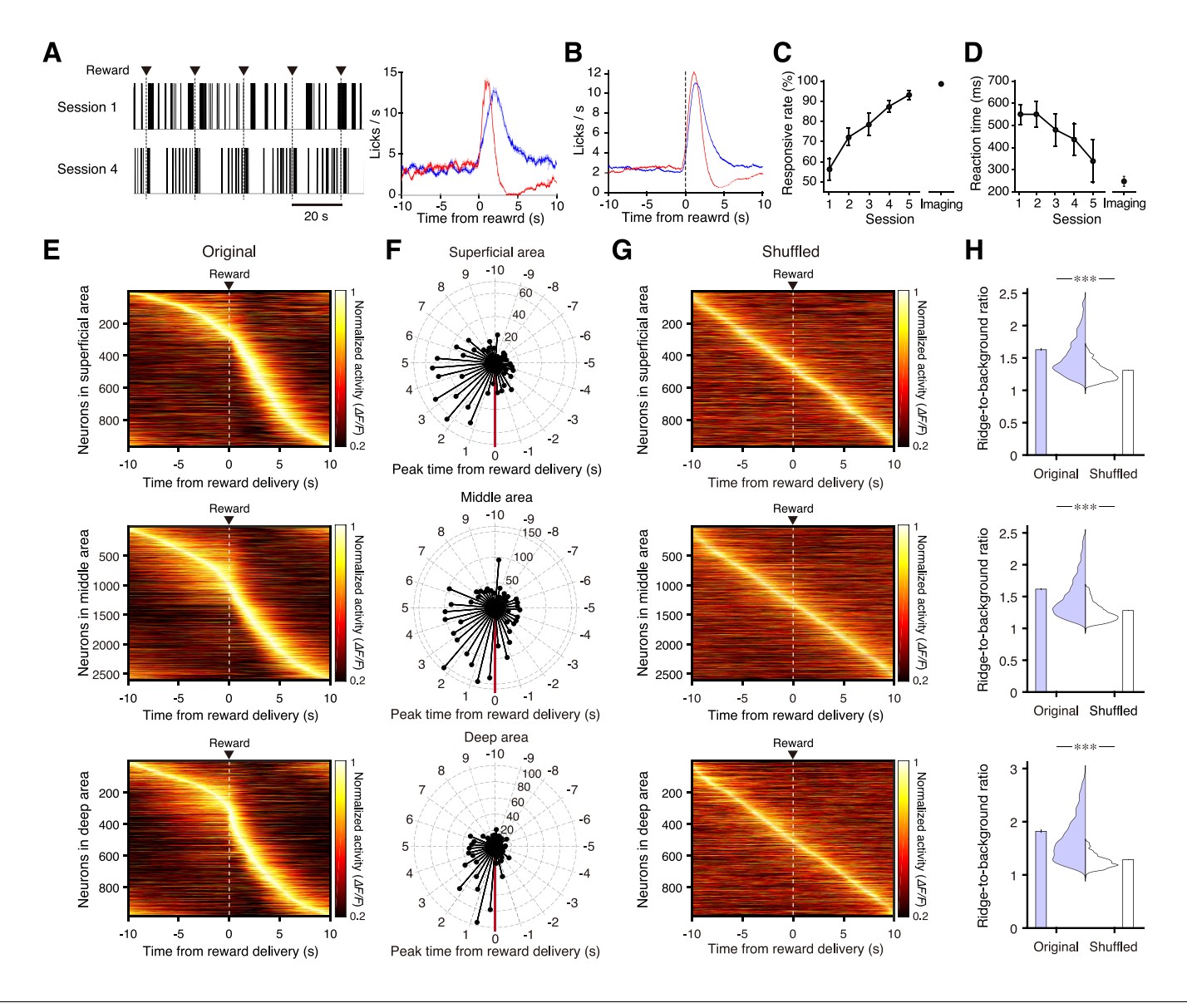

**Figure 4.** Neural activity in the medial frontal cortex during conditioning. (**A**) Left, representative traces of licking behavior during conditioning sessions 1 (top) and 4 (bottom) from the same mouse. Lines indicate the timing of licking detected by cut-off of light to an infrared LED sensor. Dashed vertical lines indicate the timing of water delivery. Right, mean licking frequency aligned with water delivery for sessions 1 (red) and 4 (blue) from the same mouse as in the left panel. Light shading indicates the s.e.m. (**B**) Mean licking frequency for sessions 1 (red) and 4 (blue; n = 11 mice). Light shading indicates the s.e.m. (**C**) Response rate in conditioning sessions 1–5 before imaging started (a response was deemed successful when licking occurred within 2 s of water delivery; n = 11 at sessions 1–4 and n = 5 at session 5) and response rate averaged over imaging blocks (n = 11 imaging blocks pooled from conditioning sessions 4–9). (**D**) Reaction time (time from the water delivery to the first lick) for conditioning sessions 1–5 before imaging started and averaged over the imaging blocks. (**E**) Normalized trial-averaged activity of each neuron aligned with water delivery (dashed lines) and ordered by the time of peak activity. Top, middle, and bottom panels are the superficial (12 fields from 7 mice), middle (35 fields from 10 mice), and deep (15 fields from 7 mice) areas, respectively. (**F**) Polar histograms of the time of peak activity from reward delivery (red line). The time bin is 0.5 s, and they are ordered clockwise from the top (−10 s to 10 s). (**G**) Normalized trial-averaged shuffled activity of each neuron. The shuffled activity was calculated by circular shifts of the original calcium traces in each trial. (**H**) Distribution and mean of the ridge-to-background ratios in original and shuffled data. Top to bottom rows correspond to the superficial (p=4.8 × 10$^{-101}$, n = 966 neurons, Wilcoxon rank-sum test), middle (p=4.3 × 10$^{-248}$, n = 2612 neurons), and deep areas (p=2.0 × 10$^{-146}$, n = 983 neurons). ***: p<0.001.

DOI: https://doi.org/10.7554/eLife.26839.013

The following figure supplements are available for figure 4:

**Figure supplement 1.** Representative trial-to-trial traces of relative fluorescence change (ΔF/F) for six neurons, aligned to water delivery.

*Figure 4 continued on next page*

*Figure 4 continued*

DOI: https://doi.org/10.7554/eLife.26839.014

**Figure supplement 2.** Ridge-to-background ratios of the mFrC neurons showing peak activity during the pre-reward period.

DOI: https://doi.org/10.7554/eLife.26839.015

**Figure supplement 3.** Ridge-to-background ratios of neurons showing peak activity during 5 s windows in the pre-reward period.

DOI: https://doi.org/10.7554/eLife.26839.016

**Figure supplement 4.** Stability of trial-by-trial activity of the mFrC neurons with peak activity during the pre-reward period.

DOI: https://doi.org/10.7554/eLife.26839.017

Narishige, Tokyo, Japan). Before virus injection, a pulled glass pipette (broken and beveled to an outer diameter of 25–30 µm; Sutter Instruments, CA, USA) and a 5 µl Hamilton syringe were back-filled with mineral oil (Nacalai Tesque, Kyoto, Japan) and front-loaded with virus solution. The virus solution was then injected into the mFrC (2.7–2.8 mm anterior and 0.4 mm left of the bregma, 800–1200 µm dorsal from the cortical surface) through a craniotomy with a small diameter of <0.5 mm. To minimize background fluorescence from solution backflow through the space made by the glass capillary insertion, the axis of the glass capillary was angled 30–40° from the horizontal plane. From 100 to 200 nl of AAV solution was injected via a syringe pump at a rate of 15–20 nl/min (KDS310; KD Scientific, MA, USA). The capillary was maintained in place for more than 10 min after the injection before being slowly withdrawn. The craniotomy was then covered with silicon elastomer (quick cast, World Precision Instruments, FL, USA) and dental adhesive (Super bond). At least 3 weeks after the viral injection, the craniotomy (2 mm square for tdTomato imaging or 1.5 mm circle for GECI imaging) was conducted at the area of interest and dura mater was removed. The craniotomies for the virus injection and for the imaging window did not overlap. With an intact dura, GECI fluorescence was not detected at depths of >1000 µm in the mFrC in our system. A glass window was placed over the craniotomy and the edge was sealed with cyanoacrylate adhesive (Vetbond, 3M, MN, USA), dental resin cement, and dental adhesive. As in *Goldey et al. (2014)*, the glass window consisted of two square cover slips (No.1, 0.12–0.17 mm thickness and 3 mm square; and No.5, 0.45–0.60 mm thickness and 2 mm square; Matsunami Glass, Osaka, Japan) or two circular cover slips (No.1, 0.12–0.17 mm thickness and 2.5 mm diameter; and No.5, 0.45–0.60 mm thickness and 1.5 mm diameter; Matsunami Glass). These were glued together with UV-curing optical adhesive (NOR-61; Norland Products, NJ, USA). After the window implantation, a 250 µl saline solution containing anti-inflammatory and analgesic carprofen (6 mg/kg) was administered intraperitoneally. Mice were then returned to their cages, and imaging sessions were started after allowing at least 1 day for recovery.

## Hippocampus

The procedures for the hippocampus were mostly the same as those for the mFrC. However, when the viral solution was injected at P12–14, the head-plate was not attached, as at their age the body size was too small to allow attachment. After virus injection, an incision to perform the injection was sutured and the pups were returned to their dam and housed until head-plate attachment. Two weeks after injection, the mice were anesthetized by intraperitoneal injection of ketamine (74 mg/kg) and xylazine (10 mg/kg), the head-plate was

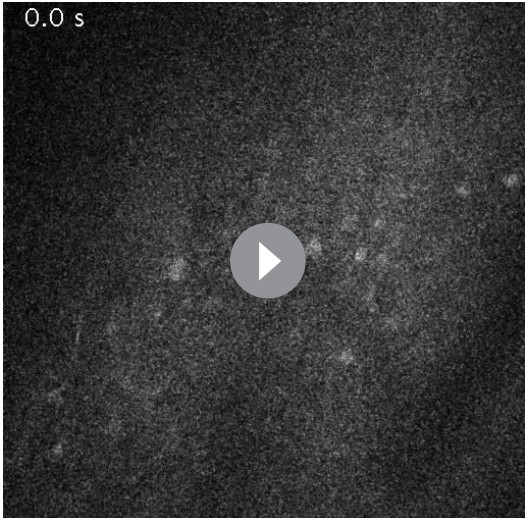

**Video 3.** Functional imaging of the PL area expressing R-CaMP1.07 during conditioning. The imaging depth was 1100 µm from the cortical surface. The field of view is 509.12 µm × 509.12 µm (512 × 512 pixels). White circles at the bottom right indicate the timing of water delivery, with an inter-delivery interval of 20 s. The frame rate was 30 Hz and the movie was downsampled to 5 Hz and denoised with a spatio-temporal Gaussian filter (spatial σ = 0.6, temporal σ = 0.8).

DOI: https://doi.org/10.7554/eLife.26839.018

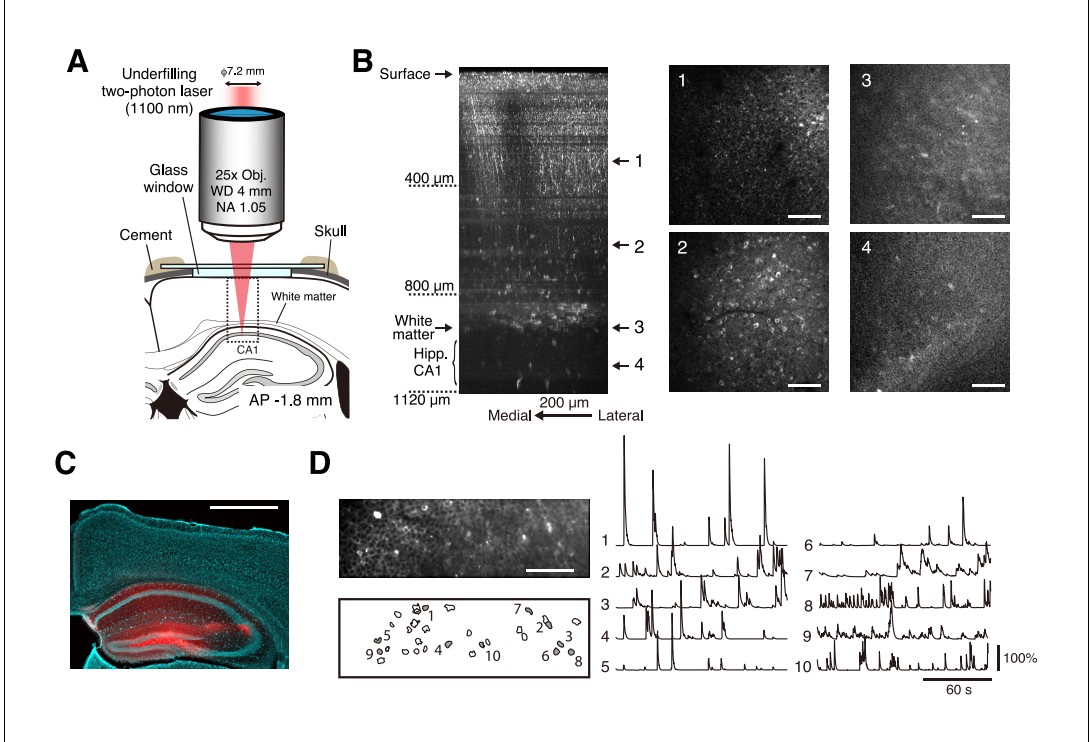

**Figure 5.** Neural activity in the hippocampal CA1 region during conditioning. (**A**) Schematic illustration of in vivo two-photon imaging of the hippocampal CA1 region. The dotted square indicates the location where the Z-stack images in (**B**) were acquired. Imaging stage was tilted so that the glass window was perpendicular to the optical axis (see Materials and methods). Immersion water was placed between the objective and the cranial window (not illustrated). (**B**) Left, representative XZ image of jRGECO1a-expressing hippocampus and deep cortical layer. The image is the maximum intensity projection of an XYZ image along Y axis. These were acquired in awake mice, and the multiple horizontal dark lines are motion artifacts. Scale bar, 200 μm. Middle and right, four XY images at the depths indicated by the number on the left. Scale bar, 200 μm. (**C**) Expression of jRGECO1a and cell nuclei after 40 min (in total) imaging of the hippocampal CA1 region. Scale bar, 1 mm. (**D**) Left, top: representative time-averaged XY image of jRGECO1a-expressing CA1 pyramidal layer at a depth of 1040 μm from the cortical surface. Scale bar, 100 μm. Dense distribution of CA1 neurons is apparent when the images are time-averaged. Left, bottom: spatial distribution of identified neurons. Middle and right, calcium transients from the corresponding numbered filled contours shown on the left. These measurements were taken while the animal was quiet, more than 15 min after the head-fixation.

DOI: https://doi.org/10.7554/eLife.26839.019

The following figure supplement is available for figure 5:

**Figure supplement 1.** Expanded immunoreactivity images of the hippocampus after imaging.

DOI: https://doi.org/10.7554/eLife.26839.020

attached, and the imaging window was implanted. The glass window consisted of two circular cover slips (No.1, 0.12–0.17 mm thickness and 2.5 mm diameter; and No.3, 0.25–0.35 mm thickness and 1.5 mm diameter). The dura mater was not removed, as it was thinner and more fragile than that in the adult mice.

## Behavioral conditioning

The mice were water-deprived in their home cages and maintained at 80–85% of their normal weight throughout the experiments. During the behavioral conditioning, mice were set within a body chamber and head-fixed with custom-designed apparatus (O'Hara, Tokyo, Japan; *Hira et al., 2013*). A spout was set in front of their mouth, and a 4 μl drop of water was delivered from the spout at a time interval of 20 s. The mice were allowed to lick at any time, and licking behavior was monitored by an infrared LED sensor. The rate of water delivery that incurred at least one lick during 2 s after the delivery was defined as the responsive rate. The duration of the daily conditioning sessions was 40–60 min. At the end of each session, the mice were allowed to freely gain water drops (total water

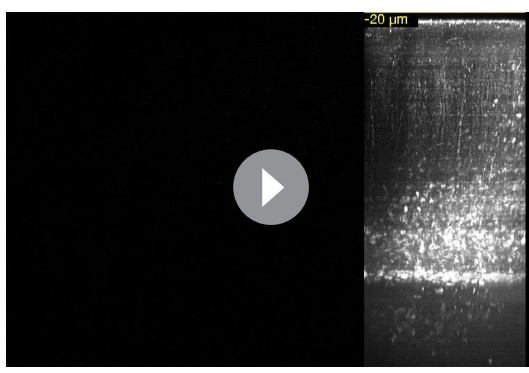

**Video 4.** Representative two-photon XYZ images of neocortex and hippocampus expressing jRGECO1a. The depth increment in the image stack was 2.5 μm and the lowest imaging depth was 1100 μm. The field of view is the same as in *Video 2*. Each image is an average of 16 frames. The mouse was not anesthetized. Motion correction was not conducted. The movie was denoised with a spatial Gaussian filter (σ = 0.6). The right image corresponds to the XZ plane of the XYZ images (maximum intensity projection toward the Y dimension) and the horizontal yellow line indicates the current depth of the left XY image. Some leakage of the virus from the hippocampus to the neocortex during the injection procedure may have resulted in a subset of the neocortical neurons expressing jRGECO1a.

DOI: https://doi.org/10.7554/eLife.26839.022

consumption was ~1 ml per session). On rest days (typically weekends), the mice had free access to a 3% agarose block (1.2 g per day) in the cage.

## Two-photon calcium imaging

Two-photon imaging was conducted using an FVMPE-RS system (Olympus, Tokyo, Japan) equipped with a 25 × water immersion objective (for imaging of the mFrC: XLPLN25XSVMP, numerical aperture: 1.00, working distance: 4 mm, Olympus; for imaging of the hippocampus: XLPLN25XWMP2, numerical aperture: 1.05, working distance: 2 mm, Olympus) and a broadly tunable laser with a pulse width of 120 fs and a repetition rate of 80 MHz (Insight DS +Dual, Spectra Physics, CA, USA), set at a wavelength of 1100 nm. Fluorescence emissions were collected using a GaAsP photomultiplier tube (Hamamatsu Photonics, Shizuoka, Japan). To shorten the light-path length within the tissue, the back aperture of the objective was underfilled with the diameter-shortened (7.2 mm, in comparison with that of the back aperture of 14.4 mm or 15.1 mm) laser beam. When the objective (XLPLN25XSVMP) was underfilled, the effective NA was calculated to be roughly 0.5 (i.e., 1.00 × 7.2/14.4). When scanning the center of the glass window at a depth of 1.2 mm from the cortical surface, the laser was assumed to be not clipped by the glass window (0.5 < 1.33 sin ($\tan^{-1}$ [0.75/1.2])=0.70).

During the imaging experiments, the mouse head was fixed and the body was constrained within a body chamber under the microscope (OPR-GST, O'Hara; *Masamizu et al., 2014*). Before the first imaging session began for each mouse, the angle of the stage on which the mouse chamber was placed was finely adjusted to set the glass window perpendicular to the optical axis. This was accomplished by the imaging of microbeads on the surface of the glass window (*Kawakami et al., 2015*). The frame acquisition rate was 30 frames/s, with a resonant scanning mirror for the X axis and a galvanometric scanning mirror for the Y axis, the pixel dwell time was 0.067 μs, and the size of the imaging fields was generally 512 × 512 pixels (0.994 μm/pixel) or 512 × 160 pixels, with three-frame averaging to increase the signal-to-noise ratio. For the strong control condition in the immunostaining experiment, the laser wavelength was tuned to 920 nm, galvanometric scanning mirrors were used for the horizontal and vertical axes, and the pixel dwell time was 200 μs. The collection collar of the objective was adjusted so that the imaging plane was well resolved. For XYZ imaging, the collar was adjusted so that deep planes were well resolved. The laser power was gradually increased from the cortical surface to the deep imaging plane. XYZ image stacks were acquired with a resonant scanner and 16–frame averaging per XY-plane. The step size was 2.5 μm unless otherwise noted. In the comparative experiments using underfilled and overfilled objectives, the laser power at the front aperture of the objective was measured in both objective configurations, and adjusted such that it was equal at each depth of imaging in the same anesthetized mice. The transmission ratio of the overfilled objective to the underfilled objective was approximately 1.5. The depth of the functional imaging plane was up to 1200 μm from the cortical surface (n = 62 planes in the mFrC from 11 mice expressing R-CaMP1.07, n = 6 in the hippocampus from three mice expressing jRGECO1a). The duration of one imaging session was 15–20 min unless otherwise noted, and 1–4 imaging sessions from different depths were performed in a daily experiment. For each mouse, imaging was conducted for 1–5 days.

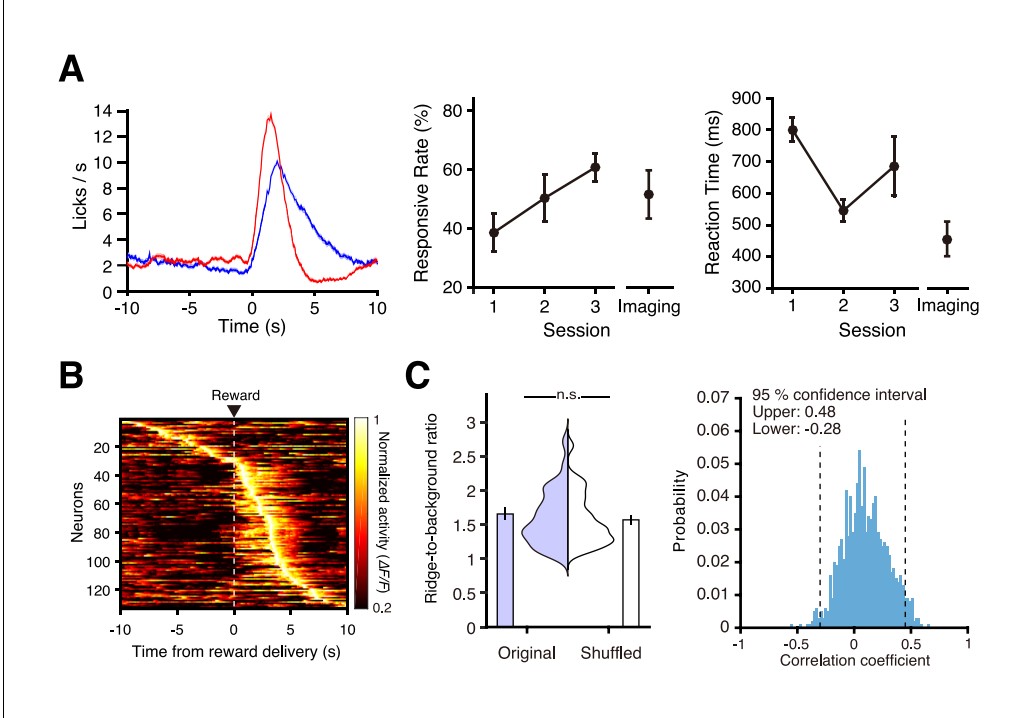

**Figure 6.** Neural activity in the hippocampal CA1 region during conditioning. (**A**) Left, mean licking frequency in sessions 1 (red) and 3 (blue; *n* = 3 mice). Light shading indicates the s.e.m. Middle, response rate in conditioning sessions 1–3 before imaging started (*n* = 3 at sessions 1–3) and response rate averaged over the imaging blocks (*n* = 6 imaging blocks pooled from conditioning session 4). Right, reaction time in conditioning sessions 1–3 before imaging started and reaction time averaged over imaging blocks. (**B**) Normalized trial-averaged activity of each neuron aligned with the water delivery (dashed lines) and ordered by the time of peak activity (six fields from three mice). (**C**) Distribution and mean of the ridge-to-background ratio of neurons with peak activity during the pre-reward period in original and shuffled data. p=0.28, *n* = 31 neurons, Wilcoxon rank-sum test. (**D**) Histograms of the correlation coefficients of the time of peak activity between the two randomly divided groups of trials in neurons with peak activity during the pre-reward period.

DOI: https://doi.org/10.7554/eLife.26839.021

## Image processing

Analyses were performed using MATLAB (R2016a, version 9.0.0.341360; MathWorks, MA, USA, RRID:SCR_001622) and Fiji software (*Schindelin et al., 2012*, RRID:SCR_002285, http://imagej.net/Fiji). Raw image sequences acquired on the FVMPE-RS system were loaded into MATLAB using custom-written scripts (http://github.com/YR-T/oir2stdData; copy archived at https://github.com/elifesciences-publications/oir2stdData). To estimate the difference between images obtained with underfilled and overfilled objectives in tdTomato-expressing animals, we calculated the bright fluorescent signal and extraction of geometric characteristics from the raw image at each depth. The bright fluorescent signal in each imaging plane was defined as the average value of the brightest 0.1% pixels (*Kobat et al., 2009*). To estimate the number of fluorescent circular structures,

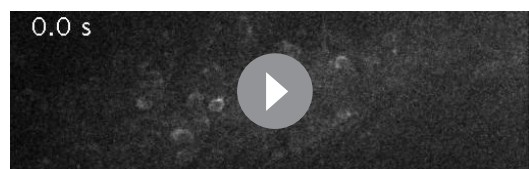

**Video 5.** Functional imaging of the hippocampal CA1 pyramidal layer expressing jRGECO1a during conditioning. The imaging depth was 1000 μm from the cortical surface and the upper cortical tissue was intact. The field of view is 509.12 μm × 159.04 μm (512 × 160 pixels). White circles at the bottom right indicate the timing of water delivery, with an inter-delivery interval of 20 s. The frame rate was 90 Hz, three frames were averaged in real time, and the three-frame-averaged data were recorded at 30 Hz. The movie was downsampled to 5 Hz and denoised with a spatio-temporal Gaussian filter (spatial σ = 0.6, temporal σ = 0.8).

DOI: https://doi.org/10.7554/eLife.26839.023

Hough transform-based detection of circles using a built-in MATLAB function (*imfindcircles* provided in the image processing toolbox) with a radius range of 6–12 pixels (approximately 6–12 μm) was applied to each XY plane. As described above, virus was injected into at 800–1200 μm from the cortical surface. This might explain why the number of detected circles increased as the imaging depth increased to 1000 μm (*Figure 1F*).

Motion correction for calcium imaging was performed by phase-correlation using the Suite2P package (*Pachitariu et al., 2016*, http://github.com/cortex-lab/Suite2P). After the motion correction, images were three frame-averaged before being analyzed. A constrained non-negative matrix factorization (cNMF) algorithm was employed to extract neural activities from a time series of images (*Pnevmatikakis et al., 2016*, http://github.com/epnev/ca_source_extraction). Then, extracted active components with soma-like contours were selected and those with dendrite- or axon-like contours were removed via visual inspection. The number of extracted active components during the conditioning experiment was as follows: 80.50 ± 26.29 (mean ± s.d., $n$ = 12 fields) in the superficial area of the mFrC, 74.66 ± 14.89 ($n$ = 35 fields) in the middle area of the mFrC, 65.53 ± 27.64 ($n$ = 15 fields) in the deep area of the mFrC, and 22.17 ± 16.32 ($n$ = 6 fields) in the hippocampal CA1 region. The noise variances in the power spectrum density at high frequency estimated by the cNMF algorithm were as follows (mean ± s.d.): 14.42 ± 5.11 ($n$ = 12 fields) in the superficial area of the mFrC, 22.00 ± 10.27 ($n$ = 35 fields) in the middle area of the mFrC, 21.69 ± 12.63 ($n$ = 15 fields) in the deep area of the mFrC, and 19.45 ± 3.68 ($n$ = 6 fields) in the hippocampal CA1 region. The detrended relative fluorescence changes ($\Delta F/F$) were calculated with eight percentile values over an interval of ±30 s around each sample time point (*Dombeck et al., 2007*). Traces of $\Delta F/F$ from 10 s before to 10 s after the water delivery in those deliveries with at least one lick during 2 s after the delivery were used for the analyses.

## Data analysis

The ridge-to-background ratio was used for the estimation of the distribution of the time of peak activity (*Harvey et al., 2012*). To create a shuffled $\Delta F/F$ trace of each neuron, the time point of the actual $\Delta F/F$ trace was circularly shifted by a random amount for each trial and then trial-averaged. For each neuron, the ridge $\Delta F/F$ was defined as the mean $\Delta F/F$ over 12 frames (100 ms/frame) surrounding the time of peak activity, and the background $\Delta F/F$ was defined as the mean $\Delta F/F$ in the other data points. The ridge $\Delta F/F$ was then divided by the background $\Delta F/F$.

The trial-by-trial stability of the time of peak activity of the neurons that had their peak activity during the pre-reward period (−10 s to 0 s) was evaluated as follows: in each session, all trials were randomly divided into two groups, and the trial-averaged activity in each group was calculated for each neuron. To remove the effects of different sample sizes across the three mFrC areas and the hippocampus, 50 neurons were randomly chosen from all imaging fields in each area. The time of peak activity in one group was plotted against that in the other, and the Pearson's correlation coefficient was determined. Thus, if the timing of the peak activity of each neuron was constant across trials, the correlation coefficient should be 1. This procedure was repeated 1000 times, and the 95% confidence interval was determined for each of the areas. When the lower bound of the 95% confidence interval was above zero, it was concluded that the time of peak activity was not random across trials. To estimate the difference in the trial-by-trial stability of the time of peak activity between pairs of the three areas in the mFrC (*Figure 4—figure supplement 4C*), the mean correlation coefficients were compared using a permutation test. For each pair from the superficial, middle, and deep areas, all neurons with peak activity during the pre-reward period were randomly reassigned to one of the two areas. For each area with reassigned neurons, the correlation coefficient between the times of peak activity of the two randomly separated groups of trials was calculated, and the absolute difference of the correlation coefficients between the two areas was estimated. This procedure was repeated 10,000 times, and the distribution of the absolute differences between the two areas was determined. Following this, the statistical significance was determined according to whether or not the absolute difference in the mean correlation coefficients between the two areas with original neurons assigned (*Figure 4—figure supplement 4B*) was above the 95th percentile of the resampled distribution corrected using the Bonferroni method. The difference in the distribution of the correlation coefficients was not due to differences in animal behavior because the mean response rate and reaction time during imaging experiments were not different among the three areas (response rate, 98.98 ± 0.72% in the superficial area, 98.63 ± 0.39% in the middle area, and

98.88 ± 0.58% in the deep area (mean ± s.e.m.), p=0.88, one-way ANOVA; reaction time, 224.08 ± 40.66 ms in the superficial area, 235.46 ± 31.44 ms in the middle area, and 272.89 ± 49.28 ms in the deep area (mean ± s.e.m.), p=0.77, one-way ANOVA).

## Histology

The mice were deeply anesthetized with ketamine (74 mg/kg) and xylazine (10 mg/kg) and transcardially perfused with 40 ml of phosphate buffered saline (PBS) and 40 ml of 4% paraformaldehyde in PBS (Wako, Osaka, Japan) 16–24 hr after the last in vivo imaging session. In the experiment to assess deep-imaging-induced damage to brain tissue, mice were perfused approximately 1 month (for imaging with 920 nm laser) or 5–10 days (under other conditions) after cranial window implantation. The brains were removed and postfixed with the same fixative at 4°C for longer than 12 hr. For immunostaining, the brains were cut into coronal sections with a thickness of 50–100 μm. Slices were washed in PBS-X (0.5% triton-X in PBS) containing 10% normal goat serum, and then incubated with one of the primary antibodies (1:500 dilution of rabbit anti-GFAP, G9269, Sigma-Aldrich, MO, USA, RRID:AB_477035; 1:500 dilution of rabbit anti-Iba1, 019–19741, Wako, RRID:AB_839504; 1:400 dilution of mouse anti-HSP70/72, ADI-SPA-810-F, Enzo Life Sciences, NY, USA, RRID:AB_10616513) overnight at 4°C. Afterwards, slices were washed in PBS-X and incubated with species-appropriate Alexa Fluoro-488 conjugated secondary antibody (1:500 dilution of anti-rabbit IgG for GFAP and Iba1 antibodies; 1:500 dilution of anti-mouse IgG for HSP70/72 antibody). After staining the cell nuclei with fluorescent Nissl stain (1:200 NeuroTrace 435/455 or NeuroTrace 640/660, N21479 or N21483, Thermo Fisher Scientific, MA, USA), the slices were mounted on glass slides with Fluoromount/Plus mounting medium (Diagnostic BioSystems, CA, USA). Fluorescence images were acquired with an upright fluorescence microscope (BX53, Olympus) and a CCD camera (Retiga 2000R, Q Imaging, BC, Canada) or all-in-one fluorescence microscope (BZ-X700, Keyence, Osaka, Japan), and analyzed with Fiji software (*Schindelin et al., 2012*), RRID:SCR_002285).

To calculate the immunoreactivity of glial and heat-shock protein activation, regions of interest with approximately 1 × 1 mm covering the cortical surface and imaged depth were selected at the center of the imaging site and the mirror position in the contralateral hemisphere. Three to five slices per immunolabel were selected for the calculation in each animal. To compensate for signal dispersion in each slice, the mean fluorescence intensity on the treated side was normalized according to the mean intensity on the contralateral side for each immunolabel. The ratios of immunoreactivity in the AAV-injected hemisphere to that in the contralateral hemisphere in mice without imaging were similar to those reported in *Podgorski and Ranganathan (2016)* but were greater than one. This may be because the cranial window implantation took place only 5–10 days before the tissue fixation and because of remaining damage from AAV injection on the ipsilateral side.

## Statistics

Data are presented as mean ± s.d., and the Wilcoxon rank-sum tests, paired *t*-test, one-way ANOVA and *post-hoc* multiple comparison with Tukey-Kramer method, Spearman's correlation tests, Pearson's correlation tests, and permutation tests described above were used for statistical comparisons. Pairwise comparisons were two-tailed unless otherwise noted. Error bars in graphs represent the s. e.m. No statistical tests were run to predetermine the sample size, and blinding and randomization were not performed.

## Acknowledgements

We thank Y Hirayama for technical assistance, J Saito, YH Tanaka, and H Wake for plasmid construction, and YR Tanaka for helpful discussion and writing a Matlab function to directly deal with images in the Olympus format. We thank V Jayaraman, DS Kim, LL Looger, and K Svoboda from the GENIE Project, Janelia Research Campus, Howard Hughes Medical Institute for providing AAV1-hSyn-jRGECO1a. This work was supported by Grant-in-Aids for Scientific Research on Innovative Areas (15H01455 and 17H06309 to MM) and for Scientific Research (A) (15H02350 to MM) from the Ministry of Education, Culture, Sports, Science, and Technology, Japan, the Strategic Research Program for Brain Sciences and the program for Brain Mapping by Integrated Neurotechnologies for Disease Studies (Brain/MINDS) from Japan Agency for Medical Research and Development to MM, and a Takeda Foundation grant to MM.

## Additional information

### Funding

| Funder | Grant reference number | Author |
|---|---|---|
| Ministry of Education, Culture, Sports, Science, and Technology | Grant-in-Aids for Scientific Research on Innovative Areas (15H01455) | Masanori Matsuzaki |
| Ministry of Education, Culture, Sports, Science, and Technology | Grant-in-Aids for Scientific Research on Innovative Areas (17H06309) | Masanori Matsuzaki |
| Ministry of Education, Culture, Sports, Science, and Technology | Grant-in-Aids for Scientific Research (A) (15H02350) | Masanori Matsuzaki |
| Takeda Science Foundation | | Masanori Matsuzaki |
| Japan Agency for Medical Research and Development | The Strategic Research Program for Brain Sciences | Masanori Matsuzaki |
| Japan Agency for Medical Research and Development | The program for Brain Mapping by Integrated Neurotechnologies for Disease Studies (Brain/MINDS) | Masanori Matsuzaki |

The funders had no role in study design, data collection and interpretation, or the decision to submit the work for publication.

### Author contributions

Masashi Kondo, Conceptualization, Software, Formal analysis, Investigation, Visualization, Methodology, Writing—original draft, Writing—review and editing; Kenta Kobayashi, Masamichi Ohkura, Junichi Nakai, Resources, Writing—review and editing; Masanori Matsuzaki, Conceptualization, Supervision, Funding acquisition, Writing—original draft, Project administration, Writing—review and editing

### Author ORCIDs

Masashi Kondo (ID) http://orcid.org/0000-0001-8818-8316
Masanori Matsuzaki (ID) http://orcid.org/0000-0003-3872-4322

### Ethics

Animal experimentation: All animal experiments were approved by the Institutional Animal Care and Use Committee of The University of Tokyo, Japan (Medicine-P16-012).

### Decision letter and Author response

Decision letter https://doi.org/10.7554/eLife.26839.025
Author response https://doi.org/10.7554/eLife.26839.026

## Additional files

### Supplementary files

• Transparent reporting form
DOI: https://doi.org/10.7554/eLife.26839.024

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
