## [Decision Letter]

Thank you for submitting your article "Two-photon calcium imaging of medial prefrontal cortex and hippocampus without cortical invasion" for consideration by *eLife*. Your article has been reviewed by three peer reviewers, and the evaluation has been overseen by a Reviewing Editor and Andrew King as the Senior Editor. The following individuals involved in review of your submission have agreed to reveal their identity: Chris Schaeffer (Reviewer #3).

The Reviewers and Reviewing Editor find considerable merit in your work, considered for publication as a "Tools and Resources" article. The Reviewing Editor in particular notes the importance of imaging deep into prefrontal cortex, which cannot be accessed by a bore hole that is placed through the overlying brain areas, such as used to access the CA1 region of hippocampus through the overlying cortex. However, given the reviewer comments, your manuscript is not suitable for publication at this point, but will require additional data and analysis. These are the key points:

We appreciate the emphasis on the need for a low NA focus of the incident light as a means to have only relatively short path lengths through the tissue and the need for high NA in terms of efficient collection. Towards this point:

1) We need to see a comparison between the depth capability of the system with the under-filled objective (for low NA excitation) versus the properly filled system (for higher resolution but presumably lower penetration depth). Thus, you should compare, in the same brain, the imaging performance at depths with the objective either under-filled or properly filled. This should be done at identical post-objective powers as well as at proportionally higher powers for the higher NA with proper filling, i.e., by a factor up to (7.2/15.1)^2 ~ 4 if possible; see thermal issue below.

2) Please present a brief discussion on the trade-offs between the imaging depth with resolution when under-filling the objective. You should refer to and discuss the work of Tung et al., (2005) as prior state of the art in the use of low NA illumination for increased depth of imaging.

There is concern over the possibility of sample tissue damage as a result of the relatively high level of laser excitation. As noted by reviewer 2, the laser power utilized in this report is higher than generally described for 2-photon (Dana et al., 2016) and even 3–photon microscopy (Ouzounov et al., 2017). Thus, a more careful analysis of potential cellular damage is required. Towards this point:

1) Show higher magnification images of the imaged region, the overlying brain surface, and nearby non-illuminated regions as a control.

2) Use a marker for cell damage. May we suggest as two possibilities staining with anti-MAP2 to look for disrupted processes in neurons or with anti-GFAP to look for activated astrocytes.

3) Include a positive control, e.g. staining after prolonged irradiation at high power, to establish sensitivity to laser-induced tissue injury. This control experiment may be part of a set of experiments that involve filling the full aperture as well (see above).

Lastly, while we recognize that this manuscript is submitted as a "Tools and Resources" article, the claims that neurons in prelimbic areas, but not in hippocampus, reliably code for reward prediction should be tempered by including other possible explanations, as noted by reviewer 1.

Reviewer #1:

This manuscript reported a method that allows two-photon calcium imaging of medial prefrontal cortex (mFrc) and hippocampus without cortical invasion. The authors showed that, by using red calcium indicators, a 1100 nm laser, and an underfilled objective back aperture, neural activity could be detected at depths of 1.0–1.2mm within mFrC of adult (2-3 months) mice. The same approach was also successful in detecting calcium transients from hippocampus of young (12-40 days) mice.

Given that both the use of red indicators and longer excitation wavelengths reduces tissue scattering, imaging depth should be increased as a result. Here, the fact that the objective pupil was underfilled was emphasized as the main factor for the larger imaging depth. To support this statement, comparisons should be made with when the microscope objective is not underfilled. In other words, the authors should compare, in the same brain, the imaging performance at depth with the objective either underfilled or properly filled, at similar post-objective power. (It would also be interesting to see a properly filled objective with higher post-objective power will perform – after all, the current data indicates that there was limited heating-induced responses at 170-180 mW, so there is some room for further power increase.)

The temporal structure of neural activity in mice mFrc and hippocampus was used to support the statement that PL neurons but not hippocampal neurons reliably code for reward prediction. However, data and data analysis presented here do not preclude other possibilities (such as activity related to motion planning or changes in attentional states). More experiments and controls should be provided. For example, record the activity of the same neurons prior to the conditioning experiments, to confirm that the temporal structure is not intrinsic to mFrc neural activity independent of the reward experiments.

Panels similar to Figure 3 should be provided for Figure 4, to confirm that the conditioning experiments were indeed successful in young mice.

Reviewer #2:

The use of an under-filled objective lens has been previously discussed in Tung et al., 2004. This prior work is relevant and should be referenced. There are also trade-offs between the imaging depth with the lateral (x,y) and z (sectioning) resolution that should be mentioned when under-filling the objective lens.

The authors of this manuscript are using a laser intensity of 170-180 mW under the objective. This is a very high level of optical power and could lead to changes in neural activity. To justify this level of power the authors reference a previous study that used 300 mW intensity in a 1-mm2 scan area. That reference found a peak temperature increase of 1.8C/100 mW and that continuous illumination with powers above 250 mW induced lasting damage. Prior work by Podgorski & Ranganathanm, (2016) found changes in neuronal firing rates with illumination-induced heating of 1.0°C.

The scan area used in this manuscript was 512x512 pixels (0.904 µm/pixel), or 463x463 µm2. This is only 21% of the 1-mm2 scan area used in the previous study by Podgorski & Ranganathanm, (2016), so heating effects would be expected to be larger. It is likely that they may have exceeded an illumination-induced heating of 1.0°C, which could impact the neural firing rates and the studied neural activity related to reward prediction. The authors should measure the brain heating in their sample under their imaging conditions to make sure it does not impact their conclusions about the neural activities.

Reviewer #3:

This is a well-written manuscript describing successful 2-photon imaging of in-vivo neural activity using long excitation wavelengths and red genetically encoded calcium indicators in behaviorally relevant rodent models. Specifically, hippocampal and medial prefrontal cortical activity patterns are evaluated in conditioned mice and found, in prefrontal cortex, to be temporally correlated with a reward stimulus. The methods and results are clearly reported and the figures present the data well. While the technical aspects of this study clearly demonstrate improvements in both in-vivo imaging depth and the use of red GECIs for recording neural activity patterns, the physiologic findings are not surprising given the existing body of knowledge on activity in prefrontal cortex in reward anticipation (Otis et al., 2017; Kim et al., 2016; Pinto and Dan, 2015), and therefore may offer limited biological insight.

The primary concern with this study is insufficient evaluation of sample tissue damage potentially incurred as a result of high power laser excitation. The laser power utilized in this report is higher than generally described for 2–photon microscopy (Dana et al., 2016) and even for 3-photon based imaging into hippocampus of the intact mouse brain (Ouzounov et al., 2017), and a more careful analysis of potential cellular damage would be helpful. At a minimum, the authors should show higher magnification images of the imaged region and the overlying brain surface (the two regions most likely to show damage). Of particular interest would be images that enabled analysis of microglia morphology, a sensitive indicator of tissue injury. It is also essential that the authors include a positive control (e.g. prolonged irradiation with >200 mW?) that establishes that the methods used are sensitive enough to detect laser-induced tissue injury when it is present.

The authors should also provide a brief review of the existing literature regarding prefrontal cortex and hippocampal roles in reward prediction. If the findings of this study corroborate previous investigations of the brain response to reward stimulus, those studies should be mentioned. Furthermore, interpreting the current results in light of what has been previously reported will help highlight the strengths and technological advancements brought forth by this report.

---

## [Author Response]

We appreciate the emphasis on the need for a low NA focus of the incident light as a means to have only relatively short path lengths through the tissue and the need for high NA in terms of efficient collection. Towards this point:1) We need to see a comparison between the depth capability of the system with the under-filled objective (for low NA excitation) versus the properly filled system (for higher resolution but presumably lower penetration depth). Thus, you should compare, in the same brain, the imaging performance at depths with the objective either under-filled or properly filled. This should be done at identical post-objective powers as well as at proportionally higher powers for the higher NA with proper filling, i.e., by a factor up to (7.2/15.1)^2 ~ 4 if possible; see thermal issue below.

We performed two-photon imaging of the same XYZ volume in the medial frontal cortex (mFrC) in the same animal when the objective was overfilled and underfilled (four volumes from two mice). To compare the fluorescence intensity under identical experimental conditions, we imaged tdTomato fluorescence because it is not dependent on the brain state, but is consistently bright. We performed the imaging with identical post-objective power at the same depths in the overfilled and underfilled configurations. We found that the mean bright fluorescence signal at each depth was always higher with the underfilled objective than with the overfilled objective (Figure 1). This might be because, in the underfilled configuration, the fluorescence signal is integrated over a larger focal volume than in the overfilled configuration and includes more fluorescent neuronal structures extending along the Z axis. The number of detected circles, which was assumed to mainly reflect the number of fluorescent neuronal somata at each depth, was greater at depths of more than 600 µm in the underfilled configuration than in the overfilled configuration (Figure 1). We have added these results to the revised manuscript.

2) Please present a brief discussion on the trade-offs between the imaging depth with resolution when under-filling the objective. You should refer to and discuss the work of Tung et al., (2005) as prior state of the art in the use of low NA illumination for increased depth of imaging.

We have cited Tung et al., (2005) in the main text (Introduction). We have added a discussion about the trade-offs in the Discussion section. In particular, we discussed the disadvantages of our method as follows:

“The major disadvantage of reducing the effective NA is a decrease in spatial resolution. In this study, the axial resolution was approximately 7 µm, which is sufficient to resolve single neurons and is unlikely to cause cross talk between neurons closely located along the Z axis (Lecoq et al., 2014). However, if dendritic branches and spines and axonal branches and boutons are the imaging target, the spatial resolution may not be sufficient.”

There is concern over the possibility of sample tissue damage as a result of the relatively high level of laser excitation. As noted by reviewer 2, the laser power utilized in this report is higher than generally described for 2-photon (Dana et al., 2016) and even 3–photon microscopy (Ouzounov et al., 2017). Thus, a more careful analysis of potential cellular damage is required. Towards this point:1) Show higher magnification images of the imaged region, the overlying brain surface, and nearby non-illuminated regions as a control.

We have added higher magnification immunostaining images (anti-GFAP, anti-Iba1, and anti-HSP70/72) of the imaged regions and the overlying brain surface from the mFrC both ipsilateral and contralateral to the imaging site in red GECI-expressing mice with and without imaging (Figure 3 and Figure 3—figure supplement 1). For the hippocampal imaging experiments, we have added higher magnification immunostaining images (anti-GFAP, anti-Iba1, and anti-HSP70/72) of the imaged hippocampal region and the overlying brain surface for hemispheres both contralateral and ipsilateral to the imaging side (Figure 5—figure supplement 1).

2) Use a marker for cell damage. May we suggest as two possibilities staining with anti-MAP2 to look for disrupted processes in neurons or with anti-GFAP to look for activated astrocytes.

We used anti-GFAP as a marker for activated astrocytes and anti-Iba1 as a marker for activated microglia to estimate the damage after imaging. We also used anti-HSP70/72 as a marker for heat-activated metabolism in glial cells and neurons. We quantitatively estimated the immunostaining intensity in the mFrC under five conditions: without imaging (as a negative control), after 15-min imaging at 800–900 µm depths, after 30-min imaging at 900–1100 µm depths, and after 30-min imaging at 300–400 µm depths (as a positive control). Under the latter three conditions, a 180 mW, 1100-nm laser was used. The fifth condition was a strong positive control: 30-min slow imaging at 200–300 µm depths with a 200 mW, 920-nm laser overfilling the objective.

Immunoreactivity 16–24 h after imaging the deep area was not statistically different from that in negative control mice. By contrast, imaging at 300–400 µm depths under the same laser conditions as the deep imaging caused significantly stronger immunoreactivity of GFAP and Iba1 than either the negative control or deep imaging. The strong positive control experiment caused significantly greater immunoreactivity of all antibodies than either the negative control or deep imaging (Figure 3, Figure 3—figure supplement 2 and Figure 3—figure supplement 3, Figure 3—source data 1). These results indicate that our assays were sensitive enough to detect laser-induced tissue injury and that two-photon imaging of the deep area with 180 mW laser power did not cause any apparent injury.

In addition, spontaneous activity of neurons detected by two-photon calcium imaging at 900–1100 µm depths was not significantly different between a 5-min period at the start of imaging and a 5-min period after 20 min of continuous imaging (Figure 3—figure supplement 3). These results strongly suggest that two-photon imaging of the deep area with ~180 mW laser power, as used in this study, did not cause any apparent damage. We have added these results to the revised manuscript.

3) Include a positive control, e.g. staining after prolonged irradiation at high power, to establish sensitivity to laser-induced tissue injury. This control experiment may be part of a set of experiments that involve filling the full aperture as well (see above).

As mentioned above, we performed immunostaining after 30-min imaging at 300–400 µm depths with the same laser power (180 mW) used for imaging at 900–1100 µm depths as a positive control. This imaging caused significantly higher anti-GFAP and anti-Iba1 immunoreactivity than either the negative control or deep imaging conditions (Figure 3 and Figure 3—source data 1). As an even stronger positive control, we performed 30-min slow imaging at 200–300 µm depths with a 200 mW, 920-nm laser overfilling the objective. This imaging caused significantly more severe immunoreactivity with all antibodies than either the negative control or deep imaging (Figure 3, Figure 3—figure supplement 2, Figure 3—source data 1). These results indicate that our assays were sensitive enough to detect laser-induced tissue injury and that two-photon imaging of the deep area with ~180 mW laser power did not cause any apparent histological injury. We have added these results to the revised manuscript.

Lastly, while we recognize that this manuscript is submitted as a "Tools and Resources" article, the claims that neurons in prelimbic areas, but not in hippocampus, reliably code for reward prediction should be tempered by including other possible explanations, as noted by reviewer 1.

In addition to the valid comment by Reviewer 1, the conditioning experiments differed slightly between the mFrC and CA1 hippocampus experiments. We have tempered the claims throughout the manuscript. We used “neural activity before water delivery”, instead of “neural activity related to reward prediction”.

Reviewer #1:This manuscript reported a method that allows two-photon calcium imaging of medial prefrontal cortex (mFrc) and hippocampus without cortical invasion. The authors showed that, by using red calcium indicators, a 1100 nm laser, and an underfilled objective back aperture, neural activity could be detected at depths of 1.0–1.2mm within mFrC of adult (2-3 months) mice. The same approach was also successful in detecting calcium transients from hippocampus of young (12-40 days) mice.Given that both the use of red indicators and longer excitation wavelengths reduces tissue scattering, imaging depth should be increased as a result. Here, the fact that the objective pupil was underfilled was emphasized as the main factor for the larger imaging depth. To support this statement, comparisons should be made with when the microscope objective is not underfilled. In other words, the authors should compare, in the same brain, the imaging performance at depth with the objective either underfilled or properly filled, at similar post-objective power. (It would also be interesting to see a properly filled objective with higher post-objective power will perform – after all, the current data indicates that there was limited heating-induced responses at 170-180 mW, so there is some room for further power increase.)

We performed two-photon imaging of the same XYZ mFrC volume in the same animal with the objective overfilled and underfilled (four volumes from two mice). To compare the fluorescence intensity between the overfilled and underfilled configurations, we imaged tdTomato fluorescence because it is consistently bright. We adjusted the post-objective laser power to be equal in both configurations. We found that the mean bright fluorescence signal at each depth was always higher in the underfilled objective than in the overfilled objective (Figure 1). This might be because, in the underfilled configuration, the fluorescence signal is integrated over a larger focal volume than in the overfilled configuration and includes more fluorescent neuronal structures extending along the Z axis. The number of detected circles, which was assumed to mainly reflect the number of fluorescent neuronal somata at each depth, was greater at depths of more than 600 µm in the underfilled configuration than in the overfilled configuration (Figure 1). The number of detected circles increased as the imaging depth increased up to 1000 µm (Figure 1). This might be because the virus was injected into at depths of 800–1200 µm. We have added these results to the revised manuscript.

In our two-photon microscopy system, the maximum laser power at the front aperture of the objective underfilled with an 1100-nm laser is 180 mW. Thus, we cannot image brain tissue with the same objective overfilled with an 1100-nm laser at a power >180 mW.

The temporal structure of neural activity in mice mFrc and hippocampus was used to support the statement that PL neurons but not hippocampal neurons reliably code for reward prediction. However, data and data analysis presented here do not preclude other possibilities (such as activity related to motion planning or changes in attentional states). More experiments and controls should be provided. For example, record the activity of the same neurons prior to the conditioning experiments, to confirm that the temporal structure is not intrinsic to mFrc neural activity independent of the reward experiments.

We agree with the Reviewer’s comments. In addition, the conditioning experiments were different for the mFrC and CA1 hippocampus. We have tempered the interpretation of the activity before reward delivery throughout the manuscript and have added a discussion about what this activity in the mFrC might represent to the Discussion section.

Panels similar to Figure 3 should be provided for Figure 4, to confirm that the conditioning experiments were indeed successful in young mice.

We have added the time course of licking behaviors in young mice in which the hippocampus was imaged (Figure 6). Conditioning was successful because the response rate increased and the reaction time decreased over sessions. However, the behavioral improvement was less than that in the adult mice in which the mFrC was imaged (please compare the middle and right panels in Figure 6 and Figure 4). This might be because the conditioning period before imaging the CA1 hippocampus was shorter (3 days) than that before imaging the mFrC (4–5 days), and because the mice in the CA1 experiment were younger than the mice in the mFrC experiment. We have removed the comparison of conditioning-related neural activity.

Reviewer #2:The use of an under-filled objective lens has been previously discussed in Tung et al., 2004. This prior work is relevant and should be referenced. There are also trade-offs between the imaging depth with the lateral (x,y) and z (sectioning) resolution that should be mentioned when under-filling the objective lens.

We have cited Tung et al., (2004) in the main text (Introduction). We have added a discussion about the trade-offs to the Discussion section. In particular, we discussed the disadvantages of our method as follows:

“The major disadvantage of reducing the effective NA is a decrease in spatial resolution. In this study, the axial resolution was approximately 7 µm, which is sufficient to resolve single neurons and is unlikely to cause cross-talk between neurons closely located along the Z axis (Lecoq et al., 2014). However, if dendritic branches and spines, and axonal branches and boutons, are the imaging target, then the spatial resolution may not be sufficient.”

The authors of this manuscript are using a laser intensity of 170-180 mW under the objective. This is a very high level of optical power and could lead to changes in neural activity. To justify this level of power the authors reference a previous study that used 300 mW intensity in a 1-mm2 scan area. That reference found a peak temperature increase of 1.8C/100 mW and that continuous illumination with powers above 250 mW induced lasting damage. Prior work by Podgorski & Ranganathanm, (2016) found changes in neuronal firing rates with illumination-induced heating of 1.0°C.The scan area used in this manuscript was 512x512 pixels (0.904 µm/pixel), or 463x463 µm2. This is only 21% of the 1-mm2 scan area used in the previous study by Podgorski & Ranganathanm, (2016), so heating effects would be expected to be larger. It is likely that they may have exceeded an illumination-induced heating of 1.0°C, which could impact the neural firing rates and the studied neural activity related to reward prediction. The authors should measure the brain heating in their sample under their imaging conditions to make sure it does not impact their conclusions about the neural activities.

As instructed by the editor, we examined the possible effects on cellular functions using immunoreactivity. We used anti-GFAP (a marker for activated astrocytes) and anti-Iba1 (a marker for activated microglia) to estimate damage after imaging. We also used anti-HSP70/72 as a marker for heat-activated metabolism in glial cells and neurons. We quantitatively estimated the immunostaining intensity in the mFrC in five conditions: without imaging (as a negative control), after 15-min imaging at 800–900 µm depths, after 30-min imaging at 900–1100 µm depths, and 30-min imaging at 300–400 µm depths (as a positive control). In the latter three conditions, a 180 mW, 1100-nm laser was used. The fifth condition was a strong positive control: 30-min slow imaging at 200–300 µm depths with a 200 mW, 920-nm laser overfilling the objective.

Immunoreactivity 16–24 h after imaging the deep area was not statistically different from that in the negative control mice. By contrast, imaging at 300–400 µm depths under the same laser conditions as deep imaging caused significantly stronger immunoreactivity of GFAP and Iba1 than either the negative control or deep imaging. The strong positive control experiment caused significantly more severe immunoreactivity for all antibodies than either the negative control or deep imaging (Figure 3, Figure 3—figure supplement 2 and Figure 3—figure supplement 3, Figure 3—source data 1). These results indicate that our assays were sensitive enough to detect laser-induced tissue injury and that two-photon imaging of the deep area with 180 mW laser power did not cause any apparent injury.

In addition, spontaneous activity of neurons in the deep area detected by two-photon calcium imaging was not different between the 5-min period at the start of imaging and a 5-min period after 20 minutes of continuous imaging (Figure 3—figure supplement 3). These results strongly suggest that two-photon imaging of the deep area with ~180 mW laser power did not cause any apparent damage.

Reviewer #3:This is a well-written manuscript describing successful 2-photon imaging of in-vivo neural activity using long excitation wavelengths and red genetically encoded calcium indicators in behaviorally relevant rodent models. Specifically, hippocampal and medial prefrontal cortical activity patterns are evaluated in conditioned mice and found, in prefrontal cortex, to be temporally correlated with a reward stimulus. The methods and results are clearly reported and the figures present the data well. While the technical aspects of this study clearly demonstrate improvements in both in-vivo imaging depth and the use of red GECIs for recording neural activity patterns, the physiologic findings are not surprising given the existing body of knowledge on activity in prefrontal cortex in reward anticipation (Otis et al., 2017; Kim et al., 2016; Pinto and Dan, 2015), and therefore may offer limited biological insight.The primary concern with this study is insufficient evaluation of sample tissue damage potentially incurred as a result of high power laser excitation. The laser power utilized in this report is higher than generally described for 2–photon microscopy (Dana et al., 2016) and even for 3-photon based imaging into hippocampus of the intact mouse brain (Ouzounov et al., 2017), and a more careful analysis of potential cellular damage would be helpful. At a minimum, the authors should show higher magnification images of the imaged region and the overlying brain surface (the two regions most likely to show damage). Of particular interest would be images that enabled analysis of microglia morphology, a sensitive indicator of tissue injury. It is also essential that the authors include a positive control (e.g. prolonged irradiation with >200 mW?) that establishes that the methods used are sensitive enough to detect laser-induced tissue injury when it is present.

We have added higher magnification immunostaining images (anti-GFAP, anti-Iba1, and anti-HSP70/72) of the imaged region and the overlying brain surface from red GECI-expressing mice with and without imaging, as shown in Figure 3 and Figure 5—figure supplement 1. In addition, we quantitatively estimated the immunostaining intensity in the mFrC under five conditions: without imaging (as a negative control), after 15-min imaging in the deep area (800–900 µm depths), after 30-min imaging in the deep area (900–1100 µm depths), after 30-min imaging in the shallow area (300–400 µm depths, as a positive control). In the latter three conditions, a 180 mW, 1100-nm laser was used. The fifth condition was a strong positive control: 30-min slow imaging at 200–300 µm depths with a 200 mW, 920-nm laser overfilling the objective.

Immunoreactivity 16–24 h after the imaging in the deep area was not statistically different from that in negative control mice. By contrast, imaging at 300–400 µm depths caused significantly stronger immunoreactivity for GFAP and Iba1 than either the negative control or deep imaging, and the strong positive-control experiment caused more severe immunoreactivity for all antibodies than either the negative control or deep imaging (Figure 3, Figure 3—figure supplement 2 and Figure 3—figure supplement 3, Figure 3—source data 1). These results indicate that our assays are sensitive enough to detect laser-induced tissue injury and that two-photon imaging of the deep area with 180 mW laser power did not cause any apparent injury.

In addition, spontaneous activity of neurons detected by two-photon calcium imaging at 900–1100 µm depths was not significantly different between a 5-min period at the start of imaging and a 5-min period after 20 min of continuous imaging (Figure 3—figure supplement 3). These results strongly suggest that two-photon imaging of the deep area with ~180 mW laser power, as used in this study, did not cause any apparent damage.

The authors should also provide a brief review of the existing literature regarding prefrontal cortex and hippocampal roles in reward prediction. If the findings of this study corroborate previous investigations of the brain response to reward stimulus, those studies should be mentioned. Furthermore, interpreting the current results in light of what has been previously reported will help highlight the strengths and technological advancements brought forth by this report.

We have added a discussion about what the mFrC activity before reward delivery might represent to the Discussion section. To image the activity in the CA1 hippocampus, we had to use young mice with less myelinated white matter than adult mice. Combined with the surgery schedule, this limited the conditioning period before imaging of the CA1 hippocampus to 3 days, which was shorter than that before imaging of the mFrC (4–5 days). On reanalyzing the behavioral data, we found that the young mice used for the CA1 imaging were conditioned more slowly than the adult mice used for the mFrC imaging (please compare the middle and right panels of Figure 6 and Figure 4). Thus, we have presented the results concerning behavioral conditioning in the mFrC and CA1 hippocampus experiments separately, and have removed the comparison of conditioning-related neural activity. Instead, we have discussed the importance of two-photon calcium imaging of the mFrC and hippocampus, and a technical strategy, as follows (Discussion section):

“The hippocampus connects the mFrC through the thalamus and the entorhinal cortex (Jingji and Stephen, 2015; Varela et al., 2014) and is thought to associate spatial, temporal, and reward information, which are required for goal-directed decision-making (Wiknheiser and Schenbaum, 2016). Here, the pre-reward activity in hippocampal CA1 neurons was not stable, likely because the conditioning period in the young mice was not sufficient to form such activity. If the adult hippocampus can be imaged through a wide-field cranial window and objective, the neural activity in both areas could be imaged simultaneously. Deep and wide-field two-photon calcium imaging of the intact brain will substantially aid our understanding of the brain circuits that integrate multimodal information in decision-making.”